# Refining Counterfactual Explanations With Joint-Distribution-Informed Shapley Towards Actionable Minimality

## Abstract

Counterfactual explanations (CE) identify data points that closely resemble the observed data but produce different machine learning (ML) model outputs, offering critical insights into model decisions. Despite the diverse scenarios, goals and tasks to which they are tailored, existing CE methods often lack actionable efficiency because of unnecessary feature changes included within the explanations that are presented to users and stakeholders. We address this problem by proposing a method that minimizes the required feature changes while maintaining the validity of CE, without imposing restrictions on models or CE algorithms, whether instance- or group-based. The key innovation lies in computing a joint distribution between observed and counterfactual data and leveraging it to inform Shapley values for feature attributions (FA). We demonstrate that optimal transport (OT) effectively derives this distribution, especially when the alignment between observed and counterfactual data is unclear in used CE methods. Additionally, a counterintuitive finding is uncovered: it may be misleading to rely on a counterfactual distribution defined by the CE generation mechanism in conducting FA. Our proposed method is validated on extensive experiments across multiple datasets, showcasing its effectiveness in refining CE towards greater actionable efficiency.

## 1 Background

Explainable Artificial Intelligence (XAI) is essential for making artificial intelligence systems transparent and trustworthy (Arrieta et al., 2020; Das & Rad, 2020). Within this area, feature attributions (FA) methods, such as Shapley values (Sundararajan & Najmi, 2020; Lundberg & Lee, 2017), determine how much each input feature contributes to a machine learning (ML) model's output. This helps simplify complex models by highlighting the most influential features. For example, in a healthcare model, Shapley values can identify key factors like age and medical history, assisting clinicians in understanding the model's decisions (Ter-Minassian et al., 2023; Nohara et al., 2022). Another technique counterfactual explanations (CE) (Wachter et al., 2017; Guidotti, 2022) show how small changes in input features can lead to different outcomes. While hundreds of CE algorithms have been proposed (Guidotti, 2022; Verma et al., 2020) to date, it is hardly practical to find one single CE algorithm that suits for all user cases, due to each of them is tailored particularly for their own different scenarios, goals, and tasks. For instance, the objective in one CE algorithm can be defined as finding a single counterfactual instance for each factual instance sometimes, while at othertimes, it could be treating multiple instances as a group and seeking one or more/multiple counterfactual instances for each factual observation. In some cases, the focus of a CE algorithm could be on the entire dataset, aiming to identify global CE that indicate the direction to move the factual instances to achieve the desired model output (Rawal & Lakkaraju, 2020; Ley et al., 2022; 2023; Carrizosa et al., 2024). Yet in other scenarios, the group of factual instances is viewed as a distribution, aiming to find a counterfactual distribution that remains similar in shape to the factual distribution (You et al., 2024), and ensuring comparable costs. Besides, some CE algorithms assume differentiable models, whereas others are designed specifically for tree-based or ensemble models.

**Problem Description and Challenges** Although both FA and CE are vital for making AI models more interpretable and accountable, relying only on one of them will cause drawbacks. That is, FA

alone may not provide actionable steps as explanations, and CE alone might include unnecessary feature changes that are not practical. Therefore, we address a general and comprehensive problem in this paper, which builds on the extensive foundations established in the literature (see Appendix A). To be specific, we seek to answer the following question:

*Given a (group of) factual instance(s), how can we devise an action plan that requires the least feature modifications to achieve a desired counterfactual outcome?*

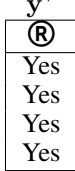

| | x | | | z′ | | | z″ | | | y* |
|---|---|---|---|---|---|---|---|---|---|---|
| 💰 | 🖱 | ® | 💰 | 🖱 | ® | 💰 | 🖱 | ® | | ® |
| 200 | 5 | No | 250 | 8 | **Yes** | 200 | 5 | No | | Yes |
| 150 | 3 | No | 150 | 3 | No | 150 | 7 | **Yes** | | Yes |
| 100 | 2 | No | 350 | 9 | **Yes** | 100 | 2 | No | | Yes |
| 150 | 6 | No | 150 | 6 | No | 350 | 6 | **Yes** | | Yes |

Figure 1: [*Example: User engagement on an e-commerce platform*] A platform aims to increase user registrations. The platform has collected data on user interactions, such as the amount of money spent (💰), the number of clicks (🖱), and whether the user has registered (®). In the original data (x), no users are registered. Action plans z′ and z″ adjust user characteristics to achieve the desired outcome (y*) of full registration. Both plans achieve a half counterfactual effect, but z″ requires fewer modifications compared to z′. This benefits customers by preserving their natural interaction patterns, leading to a better user experience. For business operators, fewer modifications result in more efficient resource allocation and cost-effective strategies, making the improvements easier to implement and more sustainable.

Three major challenges remain in addressing this problem. First, it is unrealistic to expect a single CE algorithm to meet all the needs universally, as the problem is often task-specific. Second, the approach should not rely on strong assumptions about the model (for example, requiring differentiability or special structures) to ensure its applicability across a wide range of models. Third, FA like feature importance can be misleading due to the lack of coherence between the FA scores and the changes for counterfactual effect. In other words, it is not effective to perform FA independently of CE to select the most important features to change. We will demonstrate later that this decoupling can result in counterproductive feature modifications (also referred to as actions from users/stakeholders) in Result II of Section 6, as the features deemed important generally may not align with the specific pathways to achieve the desired counterfactual outcomes.

**Main Contributions** Our main contributions are listed as follows.

- *A versatile algorithmic framework "COunterfactuals with Limited Actions (COLA)"* is proposed, which adapts to various CE methods and ML models. Extensive simulations show that the framework produces action plans that require significantly fewer feature changes to achieve outcomes similar (or sometimes equal) to those generated by various CE algorithms. Especially, COLA is shown to have near-optimal performance under certain circumstances.

- *A new Shapley method "joint-probability-informed Shapley (p-SHAP)"* is proposed, utilizing joint distribution of factual and counterfactual and resulting in remarkably well-performed action plans. We discover that other Shapley methods without incorporating counterfactual knowledge lead to unproductive attribution results for the aforementioned problem.

- *A counter-intuitive finding* is identified, showing that associating each factual data instance with its explicitly generated counterfactual (i.e. an exact alignment is known) may still underperform our p-SHAP solution. This finding emphasizes the importance of the joint distribution of factual and counterfactual instances, as a proper alignment serves as crucial knowledge for accurate contrastive FA.

To our best knowledge, this is the first method proposed for systematically addressing the problem in Figure 1 *without specifying certain CE algorithms and models.*

## 2 PROBLEM FORMULATION

We formally formulate the problem described in Figure 1. Denote $f : \mathbb{R}^d \to \mathbb{R}$ as a black-box ML model. Denote by $\mathbf{x}$ any observed (factual) data with $n$ rows and $d$ columns ($\mathbf{x} \in \mathbb{R}^{n \times d}$, $n \geq 1$, and $d \geq 1$). Let $\mathbf{y}^*$ be the target model output ($\mathbf{y}^* \in \mathbb{R}^m$, $m \geq 1$). The optimization is to look for a (group of) counterfactual data instance(s) $\mathbf{z}$ ($\mathbf{z} \in \mathbb{R}^{n \times d}$) subject to a maximum number of allowed feature changes, $C$, to achieve model output(s) as close as possible to $\mathbf{y}^*$. Let $D$ denote a divergence function that measures the dissimilarity between two entities. The problem is below.

$$\min_{\mathbf{c}, \mathbf{z}} \quad D\left(f(\mathbf{z}), \mathbf{y}^*\right) \tag{1a}$$

$$\text{s.t.} \quad D\left(\mathbf{z}, \mathbf{x}\right) \leq \epsilon \tag{1b}$$

$$\sum_{i=1}^{n} \sum_{k=1}^{d} c_{ik} \leq C \tag{1c}$$

$$z_{ik} \leq x_{ik}(1 - c_{ik}) + M c_{ik} \quad i = 1, \ldots n, \ k = 1, \ldots d \tag{1d}$$

$$z_{ik} \geq x_{ik}(1 - c_{ik}) - M c_{ik} \quad i = 1, \ldots n, \ k = 1, \ldots d \tag{1e}$$

The objective equation 1a and the constraint equation 1b formulate the typical CE optimization. Namely, $\mathbf{z}$ is expected to make $f(\mathbf{z})$ close to $\mathbf{y}^*$ yet stays close to $\mathbf{x}$. Then $\mathbf{z}$ can be used as a counterpart reference to explain why $f(\mathbf{x})$ does not achieve $\mathbf{y}^*$. We do not limit the function $D$ to any specific type of divergence function, allowing it to stay general. Example functions of $D$ can be Euclidean distance, optimal transport (OT), maximum mean discrepancy (MMD), or differences of the model outcome in mean or median. Then, equation 1c–equation 1e compose the CE optimization constrained by actions. On top of the counterfactual data $\mathbf{z}$, we also optimize an indicator variable $\mathbf{c}$, such that $z_{ik}$ is not allowed to change iff $c_{ik} = 0$. Maximum $C$ changes are allowed as imposed by equation 1c. Inspecting equation 1d and equation 1e, if $c_{ik} = 0$, $x_{ik}$ equals $z_{ik}$ and no changes happen at $(i, k)$. Otherwise, if $c_{ik} = 1$, remark that $M$ is a sufficiently large constant such that $z_{ik}$ has good freedom to change.

To solve equation 1, we resort to FA to identify the most influential features to obtain the modification indicator variable $\mathbf{c}$. The next section introduces commonly used Shapley value methods for FA, which, together with our later proposed one, are integrated into our algorithmic framework COLA, to obtain the refined counterfactual $\mathbf{z}$. The problem is computationally difficult even when $d = 1$ for linear models, see Appendix B.

## 3 PRELIMINARIES ON SHAPLEY VALUE

This section introduces commonly used Shapley value methods for FA, which, together with our later proposed one, are integrated into our algorithmic framework COLA. We first introduce the concept of Shapley value in game theory, followed by a discussion on various commonly used Shapley methods for FA. Readers could find in (Sundararajan & Najmi, 2020) for a comprehensive overview of different Shapley value methods.

In cooperative game theory, a coalitional game is characterized by a finite set of players (in our context, features), denoted by $\mathcal{F} = \{1, 2, \ldots d\}$, and a characteristic function $v$. This function $v : 2^{\mathcal{F}} \to \mathbb{R}$ maps each subset $S \subseteq \mathcal{F}$ to a real number $v(S)$, representing the total payoff that can be achieved by the members of $S$ through cooperation, with the condition $v(\emptyset) = 0$. The Shapley value, as defined based on $v$ later, is a fundamental concept in this framework, providing an equitable method to distribute the overall payoff, $v(\mathcal{F})$, of the grand coalition among the individual players.

Prior to defining the Shapley value, denote $\Delta(k, S)$ the incremental value produced by a player (feature) $k$ to a coalition $S$, aligning with the concept of marginal contribution. This is defined as the additional payoff realized by the coalition due to the inclusion of the player $k$, i.e.,

$$\Delta(k, S) = v(S \cup k) - v(S). \tag{2}$$

The Shapley value $\phi_i$ for a player $i$ in a cooperative game is defined in equation 3 below.

$$\phi_k = \frac{1}{d} \sum_{S \subset \mathcal{F} \setminus \{k\}} \binom{d-1}{|S|}^{-1} \Delta(k, S). \tag{3}$$

The value $\phi_k$ is computed by averaging the normalized marginal contributions $\Delta(k, S)$ of feature $k$ across all subsets $S$ of the feature set excluding $k$. The normalization factor is the inverse of the binomial coefficient $\binom{d-1}{|S|}$, where $d$ is the total number of features. By summing these values for all subsets and dividing by $d$, the equation provides a fair and axiomatic distribution of payoffs among players, reflecting each features's contribution among the whole collection of features.

For an arbitrary data point $\mathbf{x}_i$, we use $\mathbf{x}_{i,S}$ to represent its part containing only the features in $S$, and $\mathbf{x}_{i,\mathcal{F}\setminus S}$ the other remaining part. The same rule applies for other notations as well. Below, we introduce commonly used Shapley value methods.

*Baseline Shapley (B-SHAP)* . This approach relies on a specific baseline data point $\mathbf{r}_j$ as the reference for any $\mathbf{x}_i$. The set function is defined to be

$$v_{\mathrm{B}}^{(i)}(S) = f(\mathbf{x}_{i,S}; \mathbf{r}_{j,\mathcal{F}\setminus S}) - f(\mathbf{r}_j), \tag{4}$$

where a feature's absence is modeled using its value in the reference baseline data point $\mathbf{r}$ (Lundberg & Lee, 2017; Sun & Sundararajan, 2011; Merrick & Taly, 2020). This method assumes an exact alignment between $\mathbf{x}$ and $\mathbf{r}$.

*Random Baseline Shapley (RB-SHAP)*. This approach is a variant of BShap and is implicitly used in (Lundberg & Lee, 2017) as well as the famous "SHAP" library. The equation is as follows.

$$v_{\mathrm{RB}}^{(i)}(S) = \mathbb{E}_{\mathbf{x}'\sim\mathcal{D}}\left[f(\mathbf{x}_{i,S}; \mathbf{x}'_{\mathcal{F}\setminus S})\right] - \mathbb{E}_{\mathbf{x}'\sim\mathcal{D}}[f(\mathbf{x}')]. \tag{5}$$

where $\mathcal{D}$ is the background distribution that is commonly selected to be the training dataset (Lundberg & Lee, 2017; Merrick & Taly, 2020).

*Counterfactual Shapley (CF-SHAP)*. This is an extension from RB-SHAP, by taking the distribution $\mathcal{D}$ as the counterfactual distribution defined on every single factual instance $\mathbf{x}_i$ $(i = 1, 2, \ldots n)$.

$$v_{\mathrm{CF}}^{(i)}(S) = \mathbb{E}_{\mathbf{r}\sim\mathcal{D}(\mathbf{x}_i)}\left[f(\mathbf{x}_{i,S}; \mathbf{r}_{\mathcal{F}\setminus S})\right] - \mathbb{E}_{\mathbf{r}\sim\mathcal{D}(\mathbf{x}_i)}[f(\mathbf{r})]. \tag{6}$$

This method has been adopted in (Albini et al., 2022; Kommiya Mothilal et al., 2021), demonstrated with advantages for contrastive FA. This method assumes a probabilistic alignment between $\mathbf{x}$ and $\mathbf{r}$.

# 4 THE PROPOSED $p$-SHAP AND ITS THEORETICAL ASPECTS

**(Proposed)** $p$-**SHAP**    We generalize equation 4–equation 6 by integrating an algorithm $A_{\mathrm{Prob}}$ that returns their joint probability. Our $p$-SHAP is defined as follows.

$$v^{(i)}(S) = \mathbb{E}_{\mathbf{r}\sim p(\mathbf{r}|\mathbf{x}_i)}\left[f(\mathbf{x}_{i,S}; \mathbf{r}_{\mathcal{F}\setminus S})\right] - \mathbb{E}_{\mathbf{r}\sim p(\mathbf{r})}\left[f(\mathbf{r})\right] \tag{7a}$$

$$\text{s.t.} \quad p = A_{\mathrm{Prob}}(\mathbf{x}, \mathbf{r}) \tag{7b}$$

The reason why $p$-SHAP is a generalization of the others is as follows: First, $p$-SHAP degrades to B-SHAP in equation 4 when $A_{\mathrm{Prob}}$ defines a joint distribution between $\mathbf{x}$ and $\mathbf{r}$ that indicates an $i \leftrightarrow j$ alignment of for any $\mathbf{x}_i, \mathbf{r}_j$. Second, $p$-SHAP degrades to RB-SHAP in equation 5 when $A_{\mathrm{Prob}}$ is defined to be independent of CE but associates with an arbitrary distribution $\mathcal{D}$. Third, $p$-SHAP degrades to CF-SHAP in equation 6 when $A_{\mathrm{Prob}}$ is built upon a known distribution of CE, i.e. the data generation probability $\mathbf{r} \sim \mathcal{D}_{\mathrm{CF}}(\mathbf{x}_i)$ $(\forall i)$ is used for $A_{\mathrm{Prob}}$.

Interestingly, we emphasize that $A_{\mathrm{Prob}}$ *does not necessarily require knowledge of the CE algorithm.* The key reason that $p$-SHAP does not require explicit knowledge of how $r \sim \mathcal{D}$ is generated is that its goal is to work directly with the factual data $\mathbf{x}$, the model $f$, and the desired outcome $\mathbf{y}^*$, independent of the specific CE algorithm used to produce $\mathbf{r}$. By focusing solely on these fixed components, $p$-SHAP ensures consistency in FA without being influenced by the variability of different CE generation processes, which is a major difference to CF-SHAP. Contrary to common expectations, we demonstrate that OT can be more effective than relying on a counterfactual distribution defined by a CE generation mechanism as done by Albini et al. (2022), in Result II of Section 6 later.

Especially, one of the focus in this paper is to consider the OT problem (also the 2-Wasserstein divergence) defined below. And the transportation plan $\mathbf{p}_{\mathrm{OT}}$ obtained by solving OT is used as the joint distribution of $\mathbf{x}$ and $\mathbf{r}$ in $p$-SHAP.

$$\mathbf{p}_{\mathrm{OT}} = \arg\min_{\mathbf{p}\in\Pi(\boldsymbol{\mu},\boldsymbol{\nu})} \sum_{i=1}^{n}\sum_{j=1}^{m} p_{ij}\|\mathbf{x}_i - \mathbf{r}_j\|_2^2 + \varepsilon\sum_{i=1}^{n}\sum_{j=1}^{m} p_{ij}\log\left(\frac{p_{ij}}{\mu_i\nu_j}\right). \tag{8}$$

Note that $\boldsymbol{\mu}$ and $\boldsymbol{\nu}$ represent the marginal distributions of $\mathbf{x}$ and $\mathbf{r}$ respectively, and $\Pi(\boldsymbol{\mu}, \boldsymbol{\nu})$ the set of joint distributions (i.e. all possible transport plans). The term $\varepsilon \sum_{i=1}^{n} \sum_{j=1}^{m} p_{ij} \log(p_{ij}/(\mu_i \nu_j))$ is the entropic regularization with $\varepsilon \geq 0$ being the coefficient. Such regularization ($\varepsilon > 0$) helps accelerate the computation of OT.

**Theoretical Aspects of $p$-SHAP**   Intuitively, OT determines the most cost-effective way to move $\mathbf{x}$ closer to $\mathbf{r}$. We later prove that the transportation plan $\mathbf{p}_{\text{OT}}$, obtained by solving the OT problem, effectively guides $p$-SHAP in identifying the key features of $\mathbf{x}$ that need to be modified to bring $f(\mathbf{x})$ closer to $\mathbf{y}^*$. OT minimizes the total feature modification cost (i.e. modifying $\mathbf{x}$ towards $\mathbf{r}$) under its obtained alignment between factual $\mathbf{x}$ and counterfactual $\mathbf{r}$. This directly corresponds to our objective of finding feature modifications that achieve the counterfactual outcomes at minimal cost.

We can further strengthen this connection theoretically under the Lipschitz continuity assumption of the predictive model $f$. In Theorem 4.1 below (proof in Appendix C), we establish that the transportation plan $\mathbf{p}_{\text{OT}}$ used in $p$-SHAP is effective in minimizing an upper bound on the divergence between $f(\mathbf{x})$ and $\mathbf{y}^*$. Specifically, the 1-Wasserstein distance between $f(\mathbf{x})$ and $\mathbf{y}^*$, is bounded by the Lipschitz constant (assuming Lipschitz continuity of $f$) multiplied by the square root of the minimized expected cost of changing $\mathbf{x}$ towards $\mathbf{r}$, i.e. $\sum_{i,j} p_{ij} \|\mathbf{x}_i - \mathbf{r}_j\|_2^2$ where $p_{ij}$ ($j = 1, 2, \ldots, m$) quantify how the feature values of $\mathbf{x}_i$ should be adjusted towards those of one or multiple $\mathbf{r}_j$. Practically, this means that in $p$-SHAP, the OT plan $\mathbf{p}_{\text{OT}}$ provides a strategy to adjust the feature values of $\mathbf{x}$ towards those of $\mathbf{r}$ in a way that minimizes the expected modification cost $\sum_{i,j} p_{ij} \|\mathbf{x}_i - \mathbf{r}_j\|_2^2$. Compared to other modification plans ($\mathbf{p} \in \Pi$), $\mathbf{p}_{\text{OT}}$ yields the minimal possible cost, which in turn provides the tightest upper bound on the violation of the counterfactual effect $W_1(f(\mathbf{x}), \mathbf{y}^*)$ in proportion to this cost.

**Theorem 4.1** ($p$-SHAP Towards Counterfactual Effect). *Consider the 1-Wasserstein divergence $W_1$, i.e. $W_1(f(\mathbf{x}), \mathbf{y}^*) = \min_{\boldsymbol{\pi} \in \Pi} \sum_{i=1}^{n} \sum_{j=1}^{m} \pi_{ij} |f(\mathbf{x}_i) - \mathbf{y}_j^*|$. Suppose the counterfactual outcome $\mathbf{y}^*$ is fully achieved by $\mathbf{r}$, i.e. $\mathbf{y}_j^* = f(\mathbf{r}_j)$ ($j = 1, 2 \ldots m$). Assume that the model $f : \mathbb{R}^d \to \mathbb{R}$ is Lipschitz continuous with Lipschitz constant $L$. The expected absolute difference in model outputs between the factual and counterfactual instances, weighted by $\mathbf{p}_{OT}$ (with $\varepsilon = 0$), is bounded by:*

$$W_1(f(\mathbf{x}), \mathbf{y}^*) \leq L \sqrt{\sum_{i=1}^{n} \sum_{j=1}^{m} p_{ij}^{OT} \|\mathbf{x}_i - \mathbf{r}_j\|_2^2} \leq L \sqrt{\sum_{i=1}^{n} \sum_{j=1}^{m} p_{ij} \|\mathbf{x}_i - \mathbf{r}_j\|_2^2} \quad \forall \mathbf{p} \in \Pi.$$

*Namely, $\mathbf{p}_{OT}$ minimizes the upper bound of $W_1(f(\mathbf{x}), \mathbf{y}^*)$, where the upper bound is based on the expected feature modification cost.*

In addition, $p$-SHAP is conceptually correct in attributing the causal behavior to the modifications of the characteristics, stated in Theorem 4.2 below (proof in Appendix D).

**Theorem 4.2** (Interventional Effect of $p$-SHAP). *For any subset $S \subseteq \mathcal{F}$ and any $\mathbf{x}_i$ ($i = 1, 2, \ldots, n$), $v^{(i)}(S)$ represents the causal effect of the difference between the expected value of $f(\mathbf{r})$ under the intervention on features $S$ and the unconditional expected value of $f(\mathbf{r})$. Mathematically, this is expressed as:*

$$\mathbb{E}[f(\mathbf{r})] + v^{(i)}(S) = \mathbb{E}\left[f(\mathbf{r}) | do\left(\mathbf{r}_S = \mathbf{x}_{i,S}\right)\right].$$

Furthermore, we remark that $p$-SHAP preserves nice axioms of B-SHAP and RB-SHAP, which makes it an effective tool for attributing features. We omit the proof but refer to (Sundararajan & Najmi, 2020; Lundberg & Lee, 2017) as a reference for axioms of Shapley.

## 5   THE ALGORITHMIC FRAMEWORK COLA

**Sketch**   The algorithmic framework COLA, stated in Algorithm 1 below, aims to solve equation 1 and is established on four categories of algorithms. First, we resort to a CE algorithm $A_{\text{CE}}$ to solve the problem defined by equation 1a and equation 1b, yielding counterfactual data $\mathbf{r}$ with $\mathbf{y}^* = f(\mathbf{r})$ and $D(\mathbf{r}, \mathbf{x}) \leq \epsilon$. Second, we seek a joint distribution of $\mathbf{x}$ and $\mathbf{r}$ by an algorithm $A_{\text{Prob}}$. Third, we perform FA for $\mathbf{x}$ using a Shapley algorithm $A_{\text{Shap}}$, taking into account the joint probability. The

| Obtain $\mathbf{r}$ and compute $\mathbf{p}$ | Use $\mathbf{p}$ to compute Shapley | Match $\mathbf{x}_i$ to $\mathbf{r}_{\operatorname{argmax}_j p(\mathbf{r}_j \mid \mathbf{x}_i)}$ | Get $\mathbf{c}$ and $\mathbf{z}$ by $\boldsymbol{\varphi}$ and $\mathbf{q}$ |

Figure 2: [*An illustration of COLA*] This figure shows how COLA gets $\mathbf{c}$ and $\mathbf{z}$ for equation 1. We use $A_{\text{Value}}^{\max}$ for illustration in line 5 due to its simplicity. In lines 6–16, we assume $C = 2$, and the sampling yields exactly two positions for modfications according to the probability matrix $\boldsymbol{\varphi}$.

attributions contain the information that how large influence a row-column position would cause, if a value change happens there. Fourth, we revise $\mathbf{x}$ in $C$ positions according to the attributions, and set their values to be the ones obtained by an algorithm $A_{\text{Value}}$. This algorithm tries to ensure that the refined counterfactual $\mathbf{z}$ stay not farther away from $\mathbf{x}$ than $\mathbf{r}$. We explain Algorithm 1 in details below, along with an illustration in Figure 2.

---

**Algorithm 1** COunterfactuals with Limited Actions (COLA)

---

**Require:** Model $f$, factual $\mathbf{x} \in \mathbb{R}^{n \times d}$, target $\mathbf{y}^* \in \mathbb{R}^m$, $\epsilon$, and $C$
**Ensure:** Action plan $\mathbf{c} \in \mathbb{R}^{n \times d}$ and correspondingly a refined counterfactual $\mathbf{z} \in \mathbb{R}^{n \times d}$
  1: Use $A_{\text{CE}}(f, \mathbf{x}, \mathbf{y}^*, \epsilon)$ to obtain $\mathbf{r} \in \mathbb{R}^{m \times d}$, with $\mathbf{y}^* = f(\mathbf{r})$ and $D(\mathbf{r}, \mathbf{x}) \leq \epsilon$.
  2: Use $A_{\text{Prob}}(\mathbf{x}, \mathbf{r})$ to obtain the joint distribution matrix $\mathbf{p} \in \mathbb{R}_+^{n \times m}$.
  3: Use $A_{\text{Shap}}(\mathbf{x}, \mathbf{r}, \mathbf{p})$ to obtain the shapley values $\boldsymbol{\phi} \in \mathbb{R}^{n \times d}$ for each element of $\mathbf{x}$.
  4: Normalize the element-wise absolute values of $\boldsymbol{\phi}$, i.e., $\varphi_{ik} \leftarrow |\phi_{ik}| / \|\boldsymbol{\phi}\|_1$ ($\boldsymbol{\varphi} \in \mathbb{R}_+^{n \times d}$).
  5: Use $A_{\text{Value}}(\mathbf{r}, \mathbf{p})$ to obtain matrix $\mathbf{q} \in \mathbb{R}^{n \times d}$.
  6: For $\mathbf{c} \in \{0, 1\}^{n \times d}$, $c_{ik} \leftarrow 0$ ($i = 1 \ldots n$, $k = 1, \ldots d$).
  7: Sample $C$ pairs $(i, k)$ according to the probability matrix $\boldsymbol{\varphi}$, and let $c_{ik} = 1$ for them.
  8: Let $\mathbf{z} \leftarrow \mathbf{x}$ ($\mathbf{z} \in \mathbb{R}^{n \times d}$).
  9: **for** $i \leftarrow 1$ to $n$ **do**
 10:   **for** $k \leftarrow 1$ to $d$ **do**
 11:     **if** $c_{ik} = 1$ **then**
 12:       $z_{ik} \leftarrow q_{ik}$
 13:     **end if**
 14:   **end for**
 15: **end for**
 16: **return** $\mathbf{c}$ and $\mathbf{z}$

---

**Line 1** (*Applying a CE algorithm to find a counterfactual* $\mathbf{r}$). The CE algorithm $A_{\text{CE}}$ takes the model $f$, the factual $\mathbf{x}$, the target outcome $\mathbf{y}^*$, and the tolerance $\epsilon$ as input. The algorithm returns a counterfactual $\mathbf{r}$ staying close with $\mathbf{x}$, with $\mathbf{y}^* = f(\mathbf{r})$ and $D(\mathbf{r}, \mathbf{x}) \leq \epsilon$.

**Line 2** (*Seeking a joint distribution of* $\mathbf{x}$ *and* $\mathbf{r}$). We use an algorithm $A_{\text{Prob}}$ for this task, which takes $\mathbf{x}$ and $\mathbf{r}$ as input, and outputs a matrix representing the joint distribution of all $n$ and $m$ data points in $\mathbf{x}$ and $\mathbf{r}$, respectively. The joint distribution $\mathbf{p}$ represents an alignment relationship (or matching) between the factual rows and counterfactual rows, and we use it in Line 5 to compute the values that can be used for composing $\mathbf{z}$ later on. As discussed in Section 3, $A_{\text{Prob}}$ can be based on OT to compute a joint distribution that yields the smallest OT distance between $\mathbf{x}$ and $\mathbf{r}$, if the alignment relationship between their rows are unknown. Otherwise, it is recommended to select a joint distribution that accurately reflects the alignment between the rows in $\mathbf{x}$ and $\mathbf{r}$.

**Lines 3–4** (*p-SHAP FA*). We apply equation 7 to compute the shapley value for $\mathbf{x}$. The joint distribution $\mathbf{p}$ can be used here (without being forced) to properly align each row of $\mathbf{x}$ with its corresponding counterfactual rows of $\mathbf{r}$, such that the selected rows in $\mathbf{r}$ serve as the most representative contrastive reference for the row in $\mathbf{x}$. Numerically, this alignment significantly influences our contrastive FA.

Then, the shapley values (as a matrix) of $\mathbf{x}$ is taken element-wisely with the absolute values and normalized (such that all values sum up to one). The resulted matrix $\boldsymbol{\varphi}$ forms our FA.

**Line 5** (*Computing feature values*). The algorithm $A_{\text{Value}}$ is used for this task, which takes the counterfactual $\mathbf{r}$ and the joint distribution $\mathbf{p}$ as input. For any row $i$ in $\mathbf{x}$, $A_{\text{Value}}$ selects one or multiple row(s) in $\mathbf{r}$, used as references for making changes in $\mathbf{x}$. The algorithm returns a matrix $\mathbf{q} \in \mathbb{R}^{n \times d}$, where each element $q_{ik}$ serves as a counterfactual candidate for $x_{ik}$ ($\forall i, k$). Below, we introduce $A_{\text{Value}}^{\max}$ and $A_{\text{Value}}^{\text{avg}}$, respectively for the cases of selecting single row and selecting multiple rows.

For any row $\mathbf{x}_i$, $A_{\text{Value}}^{\max}$ selects the row of $\mathbf{r}$ with the highest probability.

$$\mathbf{q} = A_{\text{Value}}^{\max}(\mathbf{r}, \mathbf{p}) \text{ where } q_{ik} = r_{\tau(i),k} \text{ and } \tau(i) = \arg\max_{j=1,2...m} p_{ij}. \tag{9}$$

The algorithm $A^{\text{avg}}$ computes $q_{ik}$ as a convex combination as a weighted average of $r_{1k}, r_{2k}, \ldots r_{mk}$.

$$\mathbf{q} = A_{\text{Value}}^{\text{avg}}(\mathbf{r}, \mathbf{p}) \text{ where } q_{ik} = \sum_{j=1}^{m} \left( \frac{p_{ij}}{\sum_{j'=1}^{m} p_{ij'}} \right) r_{jk}. \tag{10}$$

**Lines 6–16**. Recall that the non-negative matrix $\boldsymbol{\varphi}$ is normalized to have its summation being one. We could hence treat it as a policy to select the positions in $\mathbf{x}$ for value replacement (i.e. $c_{ik} = 1$), as what line 7 does. Then, for any $i$ and $k$ with $c_{ik} = 1$, $x_{ik}$ gets modified to $q_{ik}$, and the modified matrix is then returned as $\mathbf{z}$ together with $\mathbf{c}$ forming the optimized solutions of the problem in equation 1.

By Theorem 4.1, COLA is designed to minimize the dissimilarity between $f(\mathbf{z})$ and $\mathbf{y}^*$ by modifying $\mathbf{z}$ based on feature attribution results, which identify the most important features to adjust to achieve the desired counterfactual effect. The theorem below (proof in Appendix E) demonstrates that the refined $\mathbf{z}$, produced by the COLA framework, satisfies the constraint equation 1b in the typical scenario where $n = m$, using the Frobenius norm as the distance measure. Empirical evidence supporting the general applicability of this conclusion can be found in Table 3.

**Theorem 5.1** (Counterfactual Proximity). *Let $n = m$ such that the Frobenius norm $\|\cdot\|_F$ can be used to measure the differences between $\mathbf{z}, \mathbf{r}$, and $\mathbf{x}$. Suppose that the OT plan $\mathbf{p}_{OT}$ is obtained without the entropic regularization term (i.e., $\varepsilon = 0$), resulting in a deterministic matching represented by a permutation $\sigma$ of $\{1, 2, \ldots, n\}$. Then, the refined counterfactual $\mathbf{z}$, constructed using the COLA framework, satisfies:*

$$\|\mathbf{z} - \mathbf{x}\|_F \le \|\mathbf{r} - \mathbf{x}\|_F,$$

*indicating that $\mathbf{z}$ is at least as close to $\mathbf{x}$ as $\mathbf{r}$ is, when $\mathbf{r}$ is reordered according to $\sigma$.*

**Complexity of COLA** Let $O(M_{\text{CE}})$ be the algorithm complexity of $A_{\text{CE}}$. For algorithm $A_{\text{Shap}}$, consider using weighted linear regression to estimate Shapley values, and denote by $M_{\text{Shap}}$ the number of sampled subsets. The complexity of COLA with respect to $n, m, d$, and the regularization parameter $\varepsilon$ of entropic OT is $O(M_{\text{CE}}) + O(nm \log(1/\varepsilon)) + O(nd M_{\text{Shap}}) + N$ where $N = O(nm) + O(nd)$ if $A_{\text{Value}}^{\max}$ is used and $N = O(nmd)$ if $A_{\text{Value}}^{\text{avg}}$ is used. See Appendix F.

# 6 NUMERICAL RESULTS

This section evaluates the effectiveness of COLA in addressing the problem in equation 1, with $\mathbf{y}^* = f(\mathbf{r})$ where $\mathbf{r}$ is the counterfactual obtained from a CE method $A_{\text{CE}}$. We adopt four different divergence functions: OT evaluates the distance between entire distributions. MMD evaluates the divergence between the means of two distributions in a high-dimensional feature space. The absolute mean difference (MeanD) and absolute median difference (MedianD) evaluate the divergence between mean and median, respectively. The numerical results aim at showing: I) *COLA's effectiveness for actionable minimality.* II) *p-SHAP's superior performance than other Shapley methods towards actionable minimality.* III) *COLA's near-optimal performance.*

**Experiment Setup** The experiments[1] are conducted with 4 datasets for binary classification tasks, 5 CE algorithms that are designed for diverse goals, and 12 classifiers, shown in Table 1, where a combination of dataset, $A_{\text{CE}}$ algorithm, and a model defines an "experiment scenario".

Table 1: [*Experiment Scenarios Setup*] Four datasets are used to benchmark five CE algorithms over 12 models. A "scenario" is defined to be a combination of dataset, $A_{CE}$ algorithm, and a model $f$.

| **Dataset** | HELOC (FICO, 2018), German Credit (Hofmann, 1994), Hotel Bookings (Antonio et al., 2019), COMPAS (Jeff Larson et al., 2016) |
|---|---|
| $A_{CE}$ | DiCE (Mothilal et al., 2020), AReS (Rawal & Lakkaraju, 2020), GlobeCE (Ley et al., 2023), KNN (Albini et al., 2022; Contardo et al.; Forel et al., 2023), Discount (You et al., 2024) |
| **Model** $f$ | Bagging, LightGBM, Support Vector Machine (SVM), Gaussian Process (GP), Radial Basis Function Network (RBF), XGBoost, Deep Neural Network (DNN), Random Forest (RndForest), AdaBoost, Gradient Boosting (GradBoost), Logistic Regression (LR), Quadratic Discriminant Analysis (QDA) |

Table 2: [*Experiment Methods Setup*] The table defines 6 methods for comparisons, colored to align with Figure 3. Each method is put in an experiment scenario, as defined in Table 1, for benchmarking.

| **Method** | The probability $\mathbf{p}$ used by $A_{Value}$ and $A_{Shap}$ | The Shapley algorithm $A_{Shap}$ |
|---|---|---|
| **RB-$p_{Uni}$** | $\mathbf{p} \leftarrow A_{Prob}$:$p_{ij} = 1/nm\ (\forall i, j)$ | RB-SHAP, $\mathcal{D}$ = trainset of $f$ |
| **RB-$p_{OT}$** | $\mathbf{p} \leftarrow A_{Prob}$:Eq. equation 8 (but not used in $A_{Shap}$) | RB-SHAP, $\mathcal{D}$ = trainset of $f$ |
| **CF-$p_{Uni}$** | $\mathbf{p} \leftarrow A_{Prob}$:$p_{ij} = 1/nm\ (\forall i, j)$ | CF-SHAP, $\mathcal{D}(\mathbf{x}_i) = \mathbf{p}_i$ |
| **CF-$p_{Rnd}$** | $\mathbf{p} \leftarrow A_{Prob}$:Any $\mathbf{x}_i$ matched randomly to an $\mathbf{r}_j$ | CF-SHAP, $\mathcal{D}(\mathbf{x}_i) = \mathbf{p}_i$ |
| **CF-$p_{OT}$** | $\mathbf{p} \leftarrow A_{Prob}$:Eq. equation 8 | $p$-SHAP with $\mathbf{p}$ |
| **CF-$p_{Ect}$** | $\mathbf{p} \leftarrow A_{Prob}$:Any $\mathbf{x}_i$ matched to known counterpart $\mathbf{r}_j$ | (CF or B)-SHAP, $\mathcal{D}(\mathbf{x}_i) \rightarrow \mathbf{r}_j$ |

We briefly introduce the many $A_{CE}$ in Table 1. DiCE and KNN are data-instance-based CE methods, which yield counterfactual(s) respectively for each factual instance. AReS and GlobeCE are group-based CE methods, which find a collection of counterfactual instances for the whole factual data as a group. The algorithm Discount treats the factual instances as an empirical distribution and seeks a counterfactual distribution that stays in proximity to it.

Table 2 defines 6 methods, where CF-$p_{OT}$ is the proposed $p$-SHAP and the others are baselines. Each is put in many experiment scenarios in Table 1, benchmarked comprehensively. Each method is determined by a combination of the three algorithms $A_{Prob}$, $A_{Value}$, and $A_{Shap}$. For example, the first row RB-$p_{Uni}$ uses uniform distribution as the algorithm $A_{Prob}$ to compute $\mathbf{p}$, and then the computed $\mathbf{p}$ is sequentially used in equation 9 or equation 10 of $A_{Value}$ for computing $\mathbf{q}$, and $A_{Shap}$ is RB-SHAP.

These methods are carefully designed for ablation studies. First, RB-$p_{Uni}$ differs with CF-$p_{Uni}$ in that the latter uses CE information whereas the former does not. Second, CF-$p_{Uni}$, CF-$p_{Rnd}$, and CF-$p_{OT}$ use CE in FA with different joint distributions, and we demonstrate later that the distribution computed by OT outperforms the others significantly. Third, we want to make sure that OT is useful because it informs $A_{Shap}$ with respect to the factual-counterfactual alignment, not because of other factors, and hence a comparison of CF-$p_{OT}$ that uses such an alignment to RB-$p_{OT}$ that does not. Finally, CF-$p_{Ect}$ represents a special case that each counterfactual originates from a known source, which is used as an exact factual-counterfactual alignment, making CF-SHAP also B-SHAP.

**Result I: COLA achieves significant action reduction with a minor loss in counterfactual effect**
In Table 3, we set a goal for $\mathbf{z}$ such that $f(\mathbf{z})$ reaches 80% or 100% counterfacrtual effect of $f(\mathbf{r})$[2]. Observe that COLA is effective in achieving this goal, requiring significantly fewer actions in $\mathbf{z}$ (i.e., modifications of features in $\mathbf{z}$), compared to the original CE $\mathbf{r}$. Using COLA, one could expect to only resort to 13%–25% of the feature changes (calculated by $\|\mathbf{z} - \mathbf{x}\|/\|\mathbf{r} - \mathbf{x}\|$) to achieve the counterfactual effect of 80%. In particular, only COLA with $p$-SHAP (that is, CF-$p_{OT}$) can reach the

---

[1]The code is available on both ⊙
https://anonymous.4open.science/r/Contrastive-Feature-Attribution-DFB1 and the submitted supplementary files.

[2]Note that by definition, we have $D(f(\mathbf{r}), \mathbf{y}^*) = 0$, which represents a 100% counterfactual effect since $\mathbf{y}^* = f(\mathbf{r})$. To define a counterfactual effect 80%, consider the proportion of divergence reduced by the refined CE $\mathbf{z}$. That is, *Counterfactual Effect* $= 1 - D(f(\mathbf{z}), y^*)/D(f(\mathbf{x}), y^*) = 80\%$, with $D$ being OT.

Table 3: [*COLA for Actionable Minimality*] This table shows the number of modified features in $\mathbf{z}$ by each method, when $f(\mathbf{z})$ reaches 80% counterfactual effect of $f(\mathbf{r})$. The result of each method is averaged by running in 4 randomly selected scenarios in Table 1, with 100 runs in each scenario. The symbol "-" means the target counterfactual effect cannot be achieved.

| Dataset | Method | 80% Counterfactual Effect | | 100% Counterfactual Effect | |
| --- | --- | --- | --- | --- | --- |
| | | # Modified Features | $\frac{\|\mathbf{z}-\mathbf{x}\|}{\|\mathbf{r}-\mathbf{x}\|}$ | # Modified Features | $\frac{\|\mathbf{z}-\mathbf{x}\|}{\|\mathbf{r}-\mathbf{x}\|}$ |
| German Credit $\|\mathcal{F}\|=9$ | RB-$p_{\text{Uni}}$ | – | – | – | – |
| | RB-$p_{\text{OT}}$ | $5.29(\pm 0.09)$ | $75.9\%$ | – | – |
| | CF-$p_{\text{Uni}}$ | – | – | – | – |
| | CF-$p_{\text{Rnd}}$ | – | – | – | – |
| | CF-$p_{\text{OT}}$ | $\mathbf{1.70(\pm 0.02)}$ | $\mathbf{24.3\%}$ | $\mathbf{3.13(\pm 0.03)}$ | $\mathbf{44.9\%}$ |
| Hotel Bookings $\|\mathcal{F}\|=29$ | RB-$p_{\text{Uni}}$ | $7.10(\pm 0.08)$ | $24.5\%$ | – | – |
| | RB-$p_{\text{OT}}$ | $8.55(\pm 0.08)$ | $50.2\%$ | – | – |
| | CF-$p_{\text{Uni}}$ | $7.01(\pm 0.07)$ | $41.1\%$ | – | – |
| | CF-$p_{\text{Rnd}}$ | $10.63(\pm 0.08)$ | $62.4\%$ | – | – |
| | CF-$p_{\text{OT}}$ | $\mathbf{2.50(\pm 0.03)}$ | $\mathbf{14.6\%}$ | $\mathbf{4.44(\pm 0.02)}$ | $\mathbf{26.0\%}$ |
| COMPAS $\|\mathcal{F}\|=15$ | RB-$p_{\text{Uni}}$ | $5.02(\pm 0.05)$ | $82.7\%$ | – | – |
| | RB-$p_{\text{OT}}$ | – | – | – | – |
| | CF-$p_{\text{Uni}}$ | $2.80(\pm 0.04)$ | $34.4\%$ | – | – |
| | CF-$p_{\text{Rnd}}$ | $2.58(\pm 0.04)$ | $32.1\%$ | – | – |
| | CF-$p_{\text{OT}}$ | $\mathbf{1.25(\pm 0.03)}$ | $\mathbf{14.8\%}$ | $\mathbf{2.45(\pm 0.03)}$ | $\mathbf{30.0\%}$ |
| HELOC $\|\mathcal{F}\|=23$ | RB-$p_{\text{Uni}}$ | – | – | – | – |
| | RB-$p_{\text{OT}}$ | – | – | – | – |
| | CF-$p_{\text{Uni}}$ | – | – | – | – |
| | CF-$p_{\text{Rnd}}$ | $2.73(\pm 0.04)$ | $15.7\%$ | – | – |
| | CF-$p_{\text{OT}}$ | $\mathbf{2.35(\pm 0.03)}$ | $\mathbf{13.4\%}$ | $\mathbf{7.745(\pm 0.05)}$ | $\mathbf{44.7\%}$ |

goal of the counterfactual effect of 100%, with only 26%–45% of the feature changes in original $\mathbf{r}$. Especially, $p$-SHAP leads to the best actional minimality, which is analyzed in details below.

**Result II: $p$-SHAP outperforms the other Shapley methods in achieving counterfactual effect**
We provide the evaluation of different Shapley methods in equation 4–equation 7 in Figure 3, where the x-axis is the number of allowed feature changes $C$ and the y-axis is the term $D(f(\mathbf{z}), \mathbf{y}^*)$ in equation 1. First, RB-$p_{\text{Uni}}$ and RB-$p_{\text{OT}}$ perform significantly worse than the others, indicating the importance of using CE information in attributing features towards actionable minimality in CE. Second, the result showing that CF-$p_{\text{OT}}$ outperforms RB-$p_{\text{OT}}$ demonstrates that the use of OT enhances the performance of $p$-SHAP specifically by providing effective factual-counterfactual alignment, rather than being influenced by other factors, due to that they only differ in $A_{\text{Shap}}$ (see Table 2). Third, $p$-SHAP significantly outperforms CF-$p_{\text{Uni}}$ and CF-$p_{\text{Rnd}}$. This shows the effectiveness of the joint distribution obtained in OT in using $p$-SHAP. Namely, merely using the counterfactual information for FA (as done by CF-$p_{\text{uni}}$ and CF-$p_{\text{Rnd}}$) is not enough, and a proper alignment (which does not necessarily mean the one defined by the exact counterfactual generation mechanism, as revealed in Figure 4 later) between factual and counterfactual must be considered.

We observed that COLA with $p$-SHAP possess good performance for many of the experiment scenarios defined in Table 1, as shown in Figure 3 as well as Table 3. A massive amount of such results are further demonstrated in Appendix H. In conclusion, $p$-SHAP outperforms all the other Shapley methods for the actionable minimality in CE.

**Result III: COLA may achieve near-optimal performance** This result demonstrates the effectiveness of $p$-SHAP in eliminating the influence of the CE generation process by replacing the CE algorithm-dependent knowledge[3] of $\mathcal{D}$ with the OT joint distribution between the factual and counterfactual data, shown in Figure 4. We benchmark the method CF-$p_{\text{Ect}}$ using COLA, and focus on solving equation 1 with MeanD as the divergence function $D$. Since CF-$p_{\text{Ect}}$ relies on a known

---

[3]This knowledge, i.e. exact alignment between factual and counterfactual, is available only by DiCE and KNN among all CE methods considered.

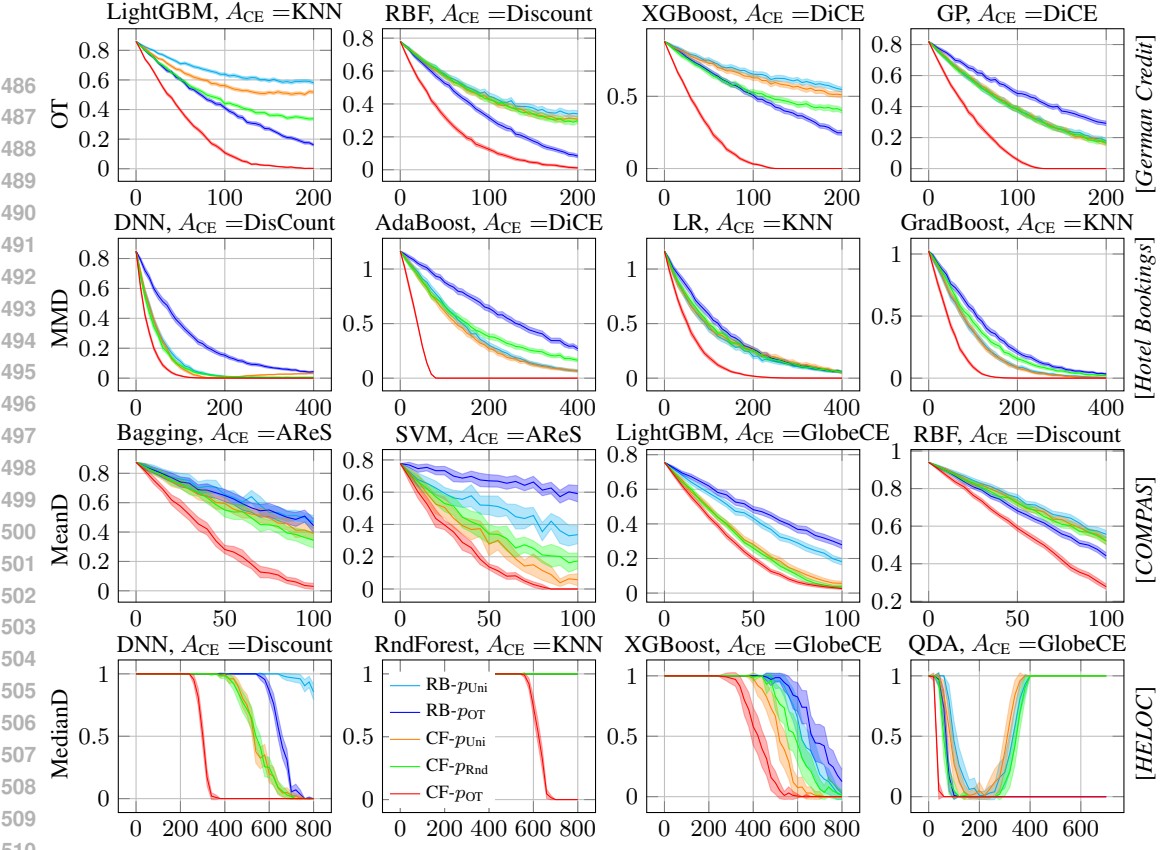

Figure 3: $D(f(\mathbf{z}), \mathbf{y}^*)$ vs. allowed actions $C$. Experiments are with 100 runs. The shadows show the 99.9% confidence intervals. $A_{\text{Value}}^{\text{avg}}$ is used for *HELOC* and *COMPAS*, and $A_{\text{Value}}^{\text{max}}$ is used for *German Credit* and *Hotel Bookings*. The legend inside "RnDForest, $A_{\text{CE}}$ =KNN" applies to all plots.

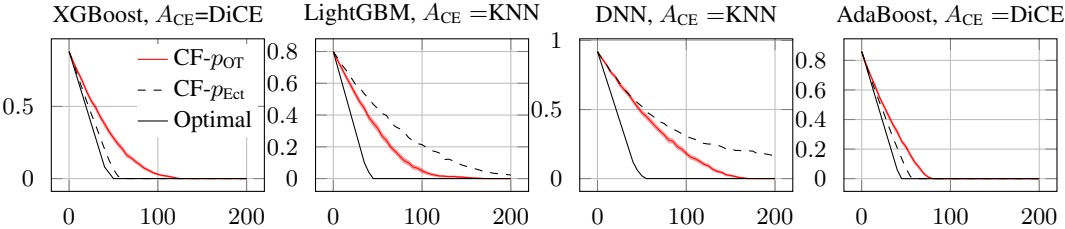

Figure 4: [*German Credit*] $D(f(z), y^*)$ vs. allowed actions $C$, with $D$ being *MeanD*.

factual-counterfactual alignment, we benchmark the effectiveness of COLA for using this alignment. The theoretical optimality of COLA in this case can be obtained by solving an mixed integer linear programming (MILP), see Appendix G for how the MILP formulation is derived. Note that solving MILP is computationally heavy, hence only done for *German Credit*.

We can see that CF-$p_{\text{Ect}}$ possess a near-optimal performance using DiCE. Remark that for DiCE and KNN, the factual-counterfactual pairs are independent to each others and hence we argue that COLA is effective for instance-based CE, even though our formulation in equation 1 is a generalization for group or distributional CE. It is interesting to note that CF-$p_{\text{OT}}$ sometimes performs better than CF-$p_{\text{Ect}}$, because it utilizes a more theoretically grounded approach to identify the key features that require modification, whereas CF-$p_{\text{Ect}}$ relies on CE algorithm-dependent knowledge, which lacks solid justification on its effectiveness for FA. We notice that there is still a gap between CF-$p_{\text{Ect}}$, CF-$p_{\text{OT}}$ and the optimal result in KNN. Finding the best alignment is still an open question.

## 7 CONCLUSIONS

This paper introduces a novel framework, COLA, for refining CE by joint-distribution-informed Shapley values, ensuring the refined CE maintains the counterfactual effect with fewer actions.

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

# APPENDICES

## A   COMPARISON TO EXISTING APPROACHES

The authors in (Albini et al., 2022) proposed CF-SHAP, which uses counterfactual data points as the background distribution for Shapley. Yet, it assumes the counterfactual data distribution is defined conditionally on each single data instance, which implies that there is a known probabilistic alignment between every factual and every counterfactual instance. Similar assumptions are made in (Merrick & Taly, 2020; Kommiya Mothilal et al., 2021). In many scenarios where global explanations are expected, this assumption fails. We note that the setup in (Albini et al., 2022) for counterfactual data distribution is a special case of ours. The authors in (Kwon & Zou, 2022) proposed WeightedSHAP, adding weights to features rather than treating them as contributing equally. Our proposed method weights the contributions for data points and can be straightforwardly extended to consider weighting both rows and columns. Literature (Kommiya Mothilal et al., 2021) establishes a framework for utilizing both FA and CE for explainability. Yet, the CE-based FA have the same assumption as (Albini et al., 2022), making it difficult to generalize to group (Ley et al., 2023; 2022) or distributional CE (You et al., 2024) cases. More importantly, the aforementioned literature does not address the minimal actions CE problem, which is the focus of our paper.

The problems formulated in (Kanamori et al., 2022; Karimi et al., 2021) are quite close to the one investigated in this paper. In (Karimi et al., 2021) the authors minimize the cost of performing actions with assumptiosn of known structural causal model (SCM), which is rarely known in practice. The authors in (Kanamori et al., 2022) pointed out that it remains open whether existing CE methods can be used for solving that problem. An-MILP-solvers based approach is proposed for linear classifiers, tree ensembles, and deep ReLU networks, built upon the works (Ustun et al., 2019; Kanamori et al., 2020; Parmentier & Vidal, 2021). However, solving MILP is costly, which makes it difficult to scale.

## B   $\mathcal{NP}$-HARDNESS OF THE PROBLEM IN EQUATION 1

Theorem B.1 below states that one does not expect a scalable exact algorithm for solving it generally, unless $\mathcal{P} = \mathcal{NP}$, and the hardness lies not only on the non-linearity of $f$, but also its combinatorial nature.

**Theorem B.1.** *Problem equation 1 is generally $\mathcal{NP}$-hard for non-trivial divergences. More specifically, it is hard even when $d = 1$ for linear models.*

*Proof.* Consider the *Sparse Regression (SR) with a Cardinality Constraint* problem, defined as follows. Given a matrix $\mathbf{W} \in \mathbb{R}^{m \times n}$, a target vector $\mathbf{y}^* \in \mathbb{R}^m$, and a sparsity level $K \in \mathbb{N}$, the goal is to find a vector $\mathbf{z} \in \mathbb{R}^n$ that minimizes the residual error while having at most $K$ non-zero

elements:

$$\min_{\mathbf{z}} \quad \|\mathbf{W}\mathbf{z} - \mathbf{y}^*\|_2^2$$
$$\text{s.t.} \quad \|\mathbf{z}\|_0 \leq K,$$

where $\|\mathbf{z}\|_0$ denotes the number of non-zero elements in $\mathbf{z}$. This problem is known to be $\mathcal{NP}$-hard due to the combinatorial nature of selecting the subset of variables to include in the model.

We will map this SR problem to our problem equation 1 with the following settings. Let the number of features be $d = 1$ and the number of instances be $n$ (the same as the dimension of the SR problem). Let the factual data be $\mathbf{x} = \mathbf{0} \in \mathbb{R}^n$ (the zero vector), and the target model output be $\mathbf{y}^* \in \mathbb{R}^m$ (as given in the SR problem). We define the model $f$ as a linear function $f(\mathbf{z}) = \mathbf{W}\mathbf{z}$.

For the model output, we define $D(f(\mathbf{z}), \mathbf{y}^*) = \|f(\mathbf{z}) - \mathbf{y}^*\|_2^2$, i.e. the Euclidean distance squared. For the instances, we define $D(\mathbf{z}, \mathbf{x}) = \|\mathbf{z} - \mathbf{x}\|_2^2 = \|\mathbf{z}\|_2^2$, since $\mathbf{x} = \mathbf{0}$. We set the maximum allowed feature changes $C$ to be equal to $k$ (the sparsity level from the SR problem). The large constant $M$ can be any sufficiently large positive number, for example, $M \geq \max_i |z_i|$.

Given that $\mathbf{x} = \mathbf{0}$, constraints equation 1d and equation 1e simplify to:

$$z_i \leq 0 \cdot (1 - c_i) + Mc_i = Mc_i,$$
$$z_i \geq 0 \cdot (1 - c_i) - Mc_i = -Mc_i, \quad \forall k = 1, \dots, n.$$

This means that if $c_i = 0$, then $z_i \leq 0$ and $z_i \geq 0$, so $z_i = 0$. If $c_i = 1$, then $z_i \in [-M, M]$. The constraint equation 1c becomes $\sum_{k=1}^{n} c_i \leq k$.

Our problem equation 1 thus becomes:

$$\min_{\mathbf{c}, \mathbf{z}} \quad \|W\mathbf{z} - \mathbf{y}^*\|_2^2 \tag{11a}$$

$$\text{s.t.} \quad \|\mathbf{z}\|_2^2 \leq \epsilon \tag{11b}$$

$$\sum_{k=1}^{d} c_i \leq k \tag{11c}$$

$$z_i = 0, \quad \text{if } c_i = 0 \tag{11d}$$

$$z_i \in [-M, M], \quad \text{if } c_i = 1 \tag{11e}$$

$$c_i \in \{0, 1\}, \quad \forall k = 1, \dots, n. \tag{11f}$$

In this formulation, the variables $c_i$ indicate whether the variable $z_i$ is allowed to change ($c_i = 1$) or not ($c_i = 0$). The constraints enforce that $z_i = 0$ when $c_i = 0$, mirroring the sparsity constraint in the SR problem. The constraint $\sum_{k=1}^{n} c_i \leq k$ ensures that at most $k$ features can change, matching the sparsity level. The objective function is identical to that of SR.

Since our problem formulation directly mirrors the SR problem with a cardinality constraint, which is known to be $\mathcal{NP}$-hard, solving Problem equation 1 is at least as hard as solving the SR problem. Therefore, Problem equation 1 is $\mathcal{NP}$-hard even when $d = 1$, the model $f$ is linear, and the divergence functions $D$ are standard Euclidean distances. $\qquad\square$

## C  PROOF OF THEOREM 4.1: $p$-SHAP TOWARDS COUNTERFACTUAL EFFECT

Let $\mathbf{x} = \{\mathbf{x}_i\}_{i=1}^{n} \in \mathbb{R}^{n \times d}$ be the set of factual data points with associated probability weights $\mu_i \geq 0$ such that $\sum_{i=1}^{n} \mu_i = 1$, and let $\mathbf{r} = \{\mathbf{r}_j\}_{j=1}^{m} \in \mathbb{R}^{m \times d}$ be the set of counterfactual data points with associated probability weights $\nu_j \geq 0$ such that $\sum_{j=1}^{m} \nu_j = 1$. Let $\mathbf{p}_{\text{OT}} \in \mathbb{R}^{n \times m}$ be the OT plan between $\mathbf{x}$ and $\mathbf{r}$ that minimizes the expected transportation cost:

$$\mathbf{p}_{\text{OT}} = \arg \min_{\mathbf{p} \in \Pi(\boldsymbol{\mu}, \boldsymbol{\nu})} \sum_{i=1}^{n} \sum_{j=1}^{m} p_{ij} \|\mathbf{x}_i - \mathbf{r}_j\|_2^2,$$

where $\Pi(\boldsymbol{\mu}, \boldsymbol{\nu})$ is the set of joint distributions satisfying the marginal constraints $\sum_{j=1}^{m} p_{ij} = \mu_i$ for all $i$ and $\sum_{i=1}^{n} p_{ij} = \nu_j$ for all $j$. The theorem below provides that feature attributions are aligned

with the expected costs of feature modifications, leading to action plans that are cost-efficient in achieving counterfactual outcomes.

**Theorem C.1** (Theorem 4.1 in the main text). *Consider the $1$-Wasserstein divergence $W_1$, i.e. $W_1(f(\mathbf{x}), \mathbf{y}^*) = \min_{\boldsymbol{\pi} \in \Pi} \sum_{i=1}^{n} \sum_{j=1}^{m} \pi_{ij} \left| f(\mathbf{x}_i) - \mathbf{y}_j^* \right|$. Suppose the counterfactual outcome $\mathbf{y}^*$ is fully achieved by $\mathbf{r}$, i.e. $\mathbf{y}_j^* = f(\mathbf{r}_j)$ $(j = 1, 2 \ldots m)$. Assume that the model $f : \mathbb{R}^d \to \mathbb{R}$ is Lipschitz continuous with Lipschitz constant $L$. The expected absolute difference in model outputs between the factual and counterfactual instances, weighted by $\mathbf{p}_{OT}$ (with $\varepsilon = 0$), is bounded by:*

$$W_1(f(\mathbf{x}), \mathbf{y}^*) \leq L\sqrt{\sum_{i=1}^{n} \sum_{j=1}^{m} p_{ij}^{OT} \|\mathbf{x}_i - \mathbf{r}_j\|_2^2} \leq L\sqrt{\sum_{i=1}^{n} \sum_{j=1}^{m} p_{ij} \|\mathbf{x}_i - \mathbf{r}_j\|_2^2} \quad \forall \mathbf{p} \in \Pi.$$

*Namely, $\mathbf{p}_{OT}$ minimizes the upper bound of $W_1(f(\mathbf{x}), \mathbf{y}^*)$, where the upper bound is based on the expected feature modification cost.*

*Proof.* Since the model $f$ is Lipschitz continuous with constant $L$, for any $\mathbf{x}_i \in \mathbb{R}^d$ and $\mathbf{r}_j \in \mathbb{R}^d$, it holds that:

$$|f(\mathbf{x}_i) - f(\mathbf{r}_j)| \leq L\|\mathbf{x}_i - \mathbf{r}_j\|_2.$$

Multiplying both sides of the inequality by $p_{ij}^{\mathrm{OT}} \geq 0$, we obtain:

$$p_{ij}^{\mathrm{OT}}|f(\mathbf{x}_i) - f(\mathbf{r}_j)| \leq L p_{ij}^{\mathrm{OT}}\|\mathbf{x}_i - \mathbf{r}_j\|_2.$$

Summing both sides over all $i = 1, \ldots, n$ and $j = 1, \ldots, m$, we have:

$$\sum_{i=1}^{n} \sum_{j=1}^{m} p_{ij}^{\mathrm{OT}}|f(\mathbf{x}_i) - f(\mathbf{r}_j)| \leq L \sum_{i=1}^{n} \sum_{j=1}^{m} p_{ij}^{\mathrm{OT}}\|\mathbf{x}_i - \mathbf{r}_j\|_2.$$

Let us denote $E_f = \sum_{i=1}^{n} \sum_{j=1}^{m} p_{ij}^{\mathrm{OT}}|f(\mathbf{x}_i) - f(\mathbf{r}_j)|$ and $E_d = \sum_{i=1}^{n} \sum_{j=1}^{m} p_{ij}^{\mathrm{OT}}\|\mathbf{x}_i - \mathbf{r}_j\|_2$. The inequality then becomes:

$$E_f \leq L E_d.$$

To further bound $E_d$, we apply the Cauchy-Schwarz inequality. Observe that the weights $p_{ij}^{\mathrm{OT}}$ are non-negative and satisfy $\sum_{i=1}^{n} \sum_{j=1}^{m} p_{ij}^{\mathrm{OT}} = 1$ because $p_{\mathrm{OT}}$ is a probability distribution over the joint space of $\mathbf{x}$ and $\mathbf{r}$. The Cauchy-Schwarz inequality states that for any real-valued functions $a_{ij}$ and $b_{ij}$,

$$\left( \sum_{i,j} a_{ij} b_{ij} \right)^2 \leq \left( \sum_{i,j} a_{ij}^2 \right) \left( \sum_{i,j} b_{ij}^2 \right).$$

Setting $a_{ij} = \sqrt{p_{ij}^{\mathrm{OT}}}$ and $b_{ij} = \sqrt{p_{ij}^{\mathrm{OT}}}\|\mathbf{x}_i - \mathbf{r}_j\|_2$, we have:

$$E_d = \sum_{i,j} p_{ij}^{\mathrm{OT}}\|\mathbf{x}_i - \mathbf{r}_j\|_2 = \sum_{i,j} \sqrt{p_{ij}^{\mathrm{OT}}}\sqrt{p_{ij}^{\mathrm{OT}}}\|\mathbf{x}_i - \mathbf{r}_j\|_2 = \sum_{i,j} a_{ij} b_{ij}.$$

Applying the Cauchy-Schwarz inequality:

$$E_d^2 \leq \left( \sum_{i,j} a_{ij}^2 \right) \left( \sum_{i,j} b_{ij}^2 \right) = \left( \sum_{i,j} p_{ij}^{\mathrm{OT}} \right) \left( \sum_{i,j} p_{ij}^{\mathrm{OT}}\|\mathbf{x}_i - \mathbf{r}_j\|_2^2 \right).$$

Since $\sum_{i,j} p_{ij}^{\mathrm{OT}} = 1$, this simplifies to:

$$E_d^2 \leq \sum_{i=1}^{n} \sum_{j=1}^{m} p_{ij}^{\mathrm{OT}}\|\mathbf{x}_i - \mathbf{r}_j\|_2^2 = \sum_{i=1}^{n} \sum_{j=1}^{m} p_{ij}^{\mathrm{OT}}\|\mathbf{x}_i - \mathbf{r}_j\|_2^2.$$

Taking the square root of both sides, we obtain:

$$E_d \leq \sqrt{\sum_{i=1}^{n} \sum_{j=1}^{m} p_{ij}^{\mathrm{OT}}\|\mathbf{x}_i - \mathbf{r}_j\|_2^2}.$$

Substituting back into the inequality for $E_f$, we have:

$$E_f \leq L \sqrt{\sum_{i=1}^{n} \sum_{j=1}^{m} p_{ij}^{\text{OT}} \|\mathbf{x}_i - \mathbf{r}_j\|_2^2}.$$

Note that the 1-Wasserstein divergence is no more than $E_f$[4], then,

$$W_1(f(\mathbf{x}), \mathbf{y}^*) = \min_{\boldsymbol{\pi} \in \Pi} \sum_{i=1}^{n} \sum_{j=1}^{m} \pi_{ij} \left| f(\mathbf{x}_i) - \mathbf{y}_j^* \right| \leq E_f \leq L \sqrt{\sum_{i=1}^{n} \sum_{j=1}^{m} p_{ij}^{\text{OT}} \|\mathbf{x}_i - \mathbf{r}_j\|_2^2}.$$

Therefore, the 1-Wasserstein divergence between $f(\mathbf{x})$ and $\mathbf{y}^*$ is bounded by the Lipschitz constant $L$ times the square root of the expected transportation cost under $\mathbf{p}_{\text{OT}}$. Since $\mathbf{p}_{\text{OT}}$ minimizes the expected transportation cost $\sum_{i,j} p_{ij} \|\mathbf{x}_i - \mathbf{r}_j\|_2^2$ over all feasible transport plans in $\Pi(\boldsymbol{\mu}, \boldsymbol{\nu})$, we have

$$\sqrt{\sum_{i=1}^{n} \sum_{j=1}^{m} p_{ij}^{\text{OT}} \|\mathbf{x}_i - \mathbf{r}_j\|_2^2} \leq \sqrt{\sum_{i=1}^{n} \sum_{j=1}^{m} p_{ij} \|\mathbf{x}_i - \mathbf{r}_j\|_2^2} \quad \forall \mathbf{p} \in \Pi(\boldsymbol{\mu}, \boldsymbol{\nu}).$$

Hence the conclusion. $\qquad\square$

## D  PROOF OF THEOREM 4.2: INTERVENTIONAL EFFECT

**Theorem D.1** (Theorem 4.2 in the main text). *For any subset $S \subseteq \mathcal{F}$ and any $\mathbf{x}_i$ ($i = 1, 2, \ldots, n$), $v^{(i)}(S)$ represents the causal effect of the difference between the expected value of $f(\mathbf{r})$ under the intervention on features $S$ and the unconditional expected value of $f(\mathbf{r})$. Mathematically, this is expressed as:*

$$\mathbb{E}[f(\mathbf{r})] + v^{(i)}(S) = \mathbb{E}\left[f(\mathbf{r}) | do\left(\mathbf{r}_S = \mathbf{x}_{i,S}\right)\right].$$

*Proof.* Let $p(\mathbf{x}, \mathbf{r})$ be the joint probability of $\mathbf{x}$ and $\mathbf{r}$ obtained from $A_{\text{Prob}}$. Under the intervention $do(\mathbf{r}_S = \mathbf{x}_{i,S})$, the features in $S$ are set to $\mathbf{x}_{i,S}$, and the features in $\mathcal{F} \backslash S$ remain distributed according to their marginal distribution $p(\mathbf{r}_{\mathcal{F} \backslash S})$. Therefore, the expected value of $f(\mathbf{r})$ under the intervention is:

$$\mathbb{E}[f(\mathbf{r}) \mid do(\mathbf{r}_S = \mathbf{x}_{i,S})] = \int_{\mathcal{R}_{\mathcal{F} \backslash S}} f(\mathbf{x}_{i,S}; \mathbf{r}_{\mathcal{F} \backslash S}) p(\mathbf{r}_{\mathcal{F} \backslash S}) \, d\mathbf{r}_{\mathcal{F} \backslash S}.$$

Remark that by the definition of $v^{(i)}(S)$,

$$v^{(i)}(S) = \mathbb{E}_{\mathbf{r} \sim p(\mathbf{r}|\mathbf{x}_i)}[f(\mathbf{x}_{i,S}; \mathbf{r}_{\mathcal{F} \backslash S})] - \mathbb{E}[f(\mathbf{r})] = \int_{\mathcal{R}_{\mathcal{F} \backslash S}} f(\mathbf{x}_{i,S}; \mathbf{r}_{\mathcal{F} \backslash S}) p(\mathbf{r}_{\mathcal{F} \backslash S}) \, d\mathbf{r}_{\mathcal{F} \backslash S} - \mathbb{E}[f(\mathbf{r})],$$

such that

$$\mathbb{E}[f(\mathbf{r})] + v^{(i)}(S) = \mathbb{E}\left[f(\mathbf{r}) | do\left(\mathbf{r}_S = \mathbf{x}_{i,S}\right)\right].$$

Hence the conclusion. $\qquad\square$

The value function $v^{(i)}(S)$ captures the expected value of $f(\mathbf{r})$ when we intervene by setting the features in $S$ to $\mathbf{x}_{i,S}$, denoted as $do(\mathbf{r}_S = \mathbf{x}_{i,S})$. This intervention is independent of any predefined joint probability distribution $p(\mathbf{x}, \mathbf{r})$. Therefore, the expression $\mathbb{E}[f(\mathbf{r})] + v^{(i)}(S)$ represents the combined effect of the base expected value of $f(\mathbf{r})$ and the additional causal impact of the attribution $v^{(i)}(S)$.

---

[4]Note that $E_f = \sum_{i=1}^{n} \sum_{j=1}^{m} p_{ij}^{\text{OT}} |f(\mathbf{x}_i) - f(\mathbf{r}_j)|$. Because both $\boldsymbol{\pi}$ and $\mathbf{p}_{\text{OT}}$ denote joint probability of $\mathbf{x}$ and $\mathbf{r}$, however, $\boldsymbol{\pi}$ makes the summation the minimum for the $W_1$ term across all possible joint distributions, whereas $\mathbf{p}_{\text{OT}}$ does not.

# E    PROOF OF THEOREM 5.1: COUNTERFACTUAL PROXIMITY

**Theorem E.1** (Theorem 5.1 in the main text). *Let $n = m$ such that the Frobenius norm $\|\cdot\|_F$ can be used to measure the differences between $\mathbf{z}$, $\mathbf{r}$, and $\mathbf{x}$. Suppose that the OT plan $\mathbf{p}_{OT}$ is obtained without the entropic regularization term (i.e., $\varepsilon = 0$), resulting in a deterministic matching represented by a permutation $\sigma$ of $\{1, 2, \ldots, n\}$. Then, the refined counterfactual $\mathbf{z}$, constructed using the COLA framework, satisfies:*

$$\|\mathbf{z} - \mathbf{x}\|_F \leq \|\mathbf{r} - \mathbf{x}\|_F,$$

*indicating that $\mathbf{z}$ is at least as close to $\mathbf{x}$ as $\mathbf{r}$ is, when $\mathbf{r}$ is reordered according to $\sigma$.*

*Proof.* Since the OT plan is computed without entropic regularization and $n = m$, the OT plan provides a deterministic matching. Specifically, for each $i$:

$$p_{ij} = \begin{cases} \frac{1}{n}, & \text{if } j = \sigma(i), \\ 0, & \text{otherwise.} \end{cases}$$

This means that each $\mathbf{x}_i$ is uniquely matched to $\mathbf{r}_{\sigma(i)}$. Note that using either $A_{\text{Value}}^{\max}$ or $A_{\text{Value}}^{\text{avg}}$, we have:

$$\mathbf{q}_i = \mathbf{r}_{\sigma(i)}, \quad \text{for all } i = 1, \ldots, n,$$

i.e., $q_{ik} = r_{\sigma(i),k}$ for all $i$ and $k$.

For the elements where $c_{ik} = 1$ (modified elements), $z_{ik} = q_{ik} = r_{\sigma(i),k}$. For elements where $c_{ik} = 0$, $z_{ik} = x_{ik}$. Therefore, we can write:

$$(z_{ik} - x_{ik})^2 = \begin{cases} (r_{\sigma(i),k} - x_{ik})^2, & \text{if } c_{ik} = 1, \\ 0, & \text{if } c_{ik} = 0. \end{cases} \tag{12}$$

The squared Frobenius norms with respect to $\mathbf{r}$ and $\mathbf{z}$ are computed as follows:

$$\|\mathbf{r} - \mathbf{x}\|_F^2 = \sum_{i=1}^{n} \sum_{k=1}^{d} (r_{\sigma(i),k} - x_{ik})^2,$$

$$\|\mathbf{z} - \mathbf{x}\|_F^2 = \sum_{i=1}^{n} \sum_{k=1}^{d} (z_{ik} - x_{ik})^2.$$

And,

$$\begin{aligned} \|\mathbf{r} - \mathbf{x}\|_F^2 - \|\mathbf{z} - \mathbf{x}\|_F^2 &= \sum_{i=1}^{n} \sum_{k=1}^{d} (r_{\sigma(i),k} - x_{ik})^2 - \sum_{i=1}^{n} \sum_{k=1}^{d} (z_{ik} - x_{ik})^2 \\ &\overset{(i)}{=} \sum_{i=1}^{n} \sum_{k=1}^{d} (r_{\sigma(i),k} - x_{ik})^2 - \sum_{(i,k):c_{ik}=1} (r_{\sigma(i),k} - x_{ik})^2 \\ &= \sum_{(i,k):c_{ik}=0} (r_{\sigma(i),k} - x_{ik})^2 \geq 0, \end{aligned}$$

where the equality (i) holds because of equation 12.

Since the difference $\|\mathbf{r} - \mathbf{x}\|_F^2 - \|\mathbf{z} - \mathbf{x}\|_F^2 \geq 0$, it follows that:

$$\|\mathbf{z} - \mathbf{x}\|_F^2 \leq \|\mathbf{r} - \mathbf{x}\|_F^2.$$

Taking square roots, we get:

$$\|\mathbf{z} - \mathbf{x}\|_F \leq \|\mathbf{r} - \mathbf{x}\|_F.$$

Hence the conclusion.    □

# F ANALYSIS OF COMPUTATIONAL COMPLEXITY OF COLA

We perform analysis of the computational complexity of COLA as follows.

First, we analyze $A_{\text{Prob}}$. If the alignment between $\mathbf{x}$ and $\mathbf{r}$ is known a priori, then $A_{\text{Prob}}$ just constructs the matrix $\mathbf{p}$ with the prior knowledge, which takes $O(n \times m)$. Otherwise, we consider solving OT to obtain $\mathbf{p}$, which, by the Sinkhorn–Knopp algorithm, takes $O(n \times m \times \log(1/\varepsilon))$.

Then we analyze $A_{\text{Shap}}$. For each subset, the model is evaluated on all $n$ data points, leading to $O(n)$ evluations per subset. Incorporating baseline values from the reference data $\mathbf{r}$ involves replacing the values of certain features with their corresponding baseline values. This operation is $O(m)$ because it requires accessing the baseline values from the reference table $\mathbf{r}$ for each of the $d$ features. If we assume that the reference values can be precomputed and accessed in constant time, then the complexity of incorporating these values can be considered as $O(d)$. The number of $M_{\text{Shap}}$ subsets results in $M_{\text{Shap}}$ model evaluations. Combining the above steps, the complexity of $A_{\text{Shap}}$ is $O(n \times d \times M_{\text{Shap}})$.

The normalization in line 4 of COLA takes $O(n \times d)$.

To compare the complexities of the two algorithms, $A_{\text{Value}}^{\text{max}}$ and $A_{\text{Value}}^{\text{avg}}$, we analyze each algorithm step-by-step. For $A_{\text{Value}}^{\text{max}}$, for each row $\mathbf{x}_i$ in the data table, we need to (1) compute the probabilities $p_{ij}$ for all $j \in \{1, 2, \ldots, m\}$, which involves $O(m)$ operations per row, (2) identify the row $\mathbf{r}_j$ in the reference data with the highest probability $p_{ij}$, which involves $O(m)$ operations per row, and (3) assign $q_{ik} = r_{\tau(i),k}$ where $\tau(i) = \arg\max_j p_{ij}$, which involves $O(d)$ operations per row. Since there are $n$ rows in the data table, the total complexity for $A_{\text{Value}}^{\text{max}}$ is $O(n \times (m + m + d)) = O(n \times (2m + d)) = O(n \times m + n \times d) = O(n \times (m + d))$.

For $A_{\text{Value}}^{\text{avg}}$, for each row $\mathbf{x}_i$ in the data table, we need to 1) compute the probabilities $p_{ij}$ for all $j \in \{1, 2, \ldots, m\}$, which involves $O(m)$ operations per row, 2) compute the sum $\sum_{j'=1}^{m} p_{ij'}$, which involves $O(m)$ operations per row, and 3) calculate the weighted average $q_{ik} = \sum_{j=1}^{m} \left( \frac{p_{ij}}{\sum_{j'=1}^{m} p_{ij'}} \right) r_{jk}$, which involves $O(m \times d)$ operations per row. Since there are $n$ rows in the data table, the total complexity for $A_{\text{Value}}^{\text{avg}}$ is $O(n \times (m + m + m \times d)) = O(n \times (2m + m \times d)) = O(n \times (m + m \times d)) = O(n \times m \times (1 + d)) = O(n \times m \times d)$.

For lines 6–16, the entire complexity is straightforwardly $O(n \times d) + O(C) = O(n \times d)$ due to the fact $C \leq n \times d$.

Therefore, the complexity of COLA using $A_{\text{Value}}^{\text{max}}$ equals

$$O(M_{\text{CE}}) + O(nm \log(1/\varepsilon)) + O(ndM_{\text{Shap}}) + O(nd) + O(n(m + d)) + O(nd)$$
$$= O(M_{\text{CE}}) + O(nm \log(1/\varepsilon)) + O(ndM_{\text{Shap}}) + O(nm) + O(nd)$$

and the complexity of COLA using $A_{\text{Value}}^{\text{avg}}$ equals

$$O(M_{\text{CE}}) + O(nm \log(1/\varepsilon)) + O(ndM_{\text{Shap}}) + O(nd) + O(nmd)) + O(nd)$$
$$= O(M_{\text{CE}}) + O(nm \log(1/\varepsilon)) + O(ndM_{\text{Shap}}) + O(nmd)$$

Hence the complexity of COLA with respect to $n$, $m$, $d$, and the regularization parameter $\varepsilon$ of entropic OT is

$$O(M_{\text{CE}}) + O(nm \log(1/\varepsilon)) + O(ndM_{\text{Shap}}) + N$$

where $N = O(nm) + O(nd)$ if $A_{\text{Value}}^{\text{max}}$ is used and $N = O(nmd)$ if $A_{\text{Value}}^{\text{avg}}$ is used.

# G AN MILP FORMULATION OF EQUATION 1 WITH MEAND

In this section, we provide a global optimality benchmark for using a known alignment between factual and counterfactual in solving equation 1 with $D$ being MeanD, namely

$$D(f(\mathbf{z}), \mathbf{y}^*) = \left| \frac{1}{n} \sum_{i=1}^{n} f(\mathbf{z}_i) - \bar{y}^* \right|$$

with $\bar{y}^* = \frac{1}{m}\sum_{j=1}^{m} y_j^*$. Since COLA is used, we have $D(\mathbf{r}, \mathbf{x}) \le \epsilon$, and $\mathbf{z}$ stays closer to $\mathbf{x}$ than $\mathbf{r}$, hence equation 1b is dropped. The formulation of equation 1 then becomes:

$$\min_{\mathbf{c},\mathbf{z}} \quad \left| \sum_{i=1}^{n} f(\mathbf{z}_i) - n\bar{y}^* \right| \tag{13a}$$

$$\text{s.t.} \quad \sum_{i=1}^{n}\sum_{k=1}^{d} c_{ik} \le C \tag{13b}$$

$$z_{ik} = r_{ik}c_{ik} + x_{ik}(1 - c_{ik}) \quad i = 1, \dots n, \; k = 1, \dots d \tag{13c}$$

Note that the original constraints equation 1d and equation 1e merge to be equation 13c, because CF-$p_{\text{Ect}}$ is imposed to be used. That is, for any $\mathbf{x}_i$, there is an exact $\mathbf{r}_j$ serves as its reference in $A_{\text{Shap}}$ and $A_{\text{Value}}$, such that $x_{ik}$ ($k = 1, 2 \dots d$) either stays unchanged or can be changed to $r_{jk}$. Therefore $z_{ik} = r_{ik}c_{ik} + x_{ik}(1 - c_{ik})$ of which the value depends on the binary variable $c_{ik}$.

Due to the known alignment between any $\mathbf{x}_i$ and its corresponding $\mathbf{r}_j$, $\mathbf{q}$ is determined (also, both $A_{\text{Values}}^{\max}$ and $A_{\text{Values}}^{\text{avg}}$ return the same $\mathbf{q}$). For any data point $i$ and any feature set $S \subseteq \mathcal{F}$, let $\mathbf{z}_{iS}$ denotes the solution $\mathbf{z}_i$ where we have all features $k \in S$ changed to $q_{ik}$, and the other features $h \in \mathcal{F}\backslash S$ stays $x_{ih}$. Hence the set of $\mathbf{z}_{iS}$ ($S \subseteq \mathcal{F}$) composes the domain of all possible values of $\mathbf{z}$. Define a corresponding scaler variable for any $\mathbf{z}_{iS}$:

$$g_{iS} = f(\mathbf{z}_{iS}) - \bar{y}^*.$$

Then, for any $\mathbf{z}_i$ in equation 13, the value of the term $\sum_{i=1}^{n} f(z_i) - n\bar{y}^*$ can be represented by a binary variable $a_{iS}$ together with the scaler $g_{iS}$, namely,

$$\sum_{i=1}^{n} f(\mathbf{z}_i) - n\bar{y}^* = \sum_{i=1}^{n}\sum_{S \subseteq \mathcal{F}} g_{iS}a_{iS}.$$

The optimization problem equation 13 is hence reformulated as a mixed integer programming below.

$$\min_{\mathbf{a},\eta} \quad \eta \tag{14a}$$

$$\text{s.t.} \quad \sum_{i=1}^{n}\sum_{S \subseteq \mathcal{F}} g_{iS}a_{iS} \le \eta \tag{14b}$$

$$\sum_{i=1}^{n}\sum_{S \subseteq \mathcal{F}} g_{iS}a_{iS} \ge -\eta \tag{14c}$$

$$\sum_{S \subseteq \mathcal{F}} a_{iS} = 1 \quad i = 1, \dots n \tag{14d}$$

$$\sum_{i=1}^{n}\sum_{S \subseteq \mathcal{F}} |S|a_{iS} \le C \tag{14e}$$

Minimizing $\eta$ under the two constraints equation 14b and equation 14c is equivalent to minimizing the objective function of equation 13. The constraints in equation 14d guarantees that each data point $i$ is subject to one and only one feature modification plan $a_{iS}$ for a specific $S$ ($S \subseteq \mathcal{F}$). The constraint equation 14e corresponds to equation 13b. Solving equation 14 yields the theoretical optimality of COLA using a known alignment between factual and counterfactual, demonstrated in Figure 4 in Section 6.

# H  EXTENDED NUMERICAL RESULTS

Observing Figures 5–8, CF-$p_{\text{Uni}}$ generally performs better than RB-$p_{\text{Uni}}$. Second, consider RB-$p_{\text{OT}}$ and CF-$p_{\text{OT}}$ that also differ only in $A_{\text{Shap}}$, the latter consistently outperforms the former. Hence *RB-SHAP is not suitable for FA in CE*.

We analyze how different Shapley methods affect FA, corresponding to lines 3–4 in COLA. The shapley methods can be classified into two categories: First, consider RB-$p_{\text{Uni}}$ and CF-$p_{\text{Uni}}$ that differ

only in $A_{\text{Shap}}$. Observing Figures 5–8, CF-$p_{\text{Uni}}$ generally performs better than RB-$p_{\text{Uni}}$. Second, consider RB-$p_{\text{OT}}$ and CF-$p_{\text{OT}}$ that also differ only in $A_{\text{Shap}}$, the latter consistently outperforms the former. Hence *RB-SHAP is not suitable for FA in CE*.

Besides FA, the other equally important step of COLA is line 5, i.e. using the joint probability $p(\mathbf{x}, \mathbf{r})$ to compose the matrix $\mathbf{q}$, telling the factual $x$ to which direction to change its features so as to move towards the target model outcome. We observe in Figures 5–8 that CF-$p_{\text{OT}}$ consistently outperforms all other methods throughout all experiments. Note that all the three methods CF-$p_{\text{Uni}}$, CF-$p_{\text{Rnd}}$, and CF-$p_{\text{OT}}$ provide solution's for the joint probability $\mathbf{p}$ when the exact alignment between factuals and counterfactuals are unknown. Yet, their performance differ significantly. Simply knowing the CE (and its marginal distribution) is insufficient.

OT proves to be exceptionally useful when the alignment information between factual and counterfactual instances is missing or inaccurate. Even when the CE algorithm explicitly matches each factual instance to a corresponding counterfactual, it is challenging to justify that the known alignment optimizes performance. This is supported by Figure 4 in Section 6.

Note that $p_{\text{OT}}$ does not need to be the true joint distribution of $\mathbf{x}$ and $\mathbf{r}$ from a data generation perspective. Instead, it should guide COLA to treat $\mathbf{x}$ and $\mathbf{r}$ together for both FA and CE. Furthermore, the QDA column in Figure 5 shows stableness of OT-based methods, while others diverge significantly from the target. We emphasize that COLA, however, is *not limited to using OT* as $A_{\text{Prob}}$. As indicated by Figure 4, any known best $\mathbf{p}$ still has non-negligible gap to the global optimality. Devising a better $A_{\text{Prob}}$ algorithm is hence an interesting topic worth exploration.

## I  EXPERIMENTS REPRODUCIBILITY

The experiments are conducted on a high performance computing (HPC) cluster, running with four nodes (for the four datasets) in parallel, with each node equipped with two Intel Xeon Processor 2660v3 (10 core, 2.60GHz) and 128 GB memory. The experiment runs approximately 5-10 hours in each node, dependent on the size of the dataset. It is also possible to reproduce the experiment on a laptop, while it costs more computational time generally than using an HPC cluster.

For the four datasets, the numerical features are standardized, and the categorical features follow either label-encoding or one-hot encoding. Practically, we did not observe remarkable difference between the two encoding methods in terms of COLA's performance. The train-test split follows $7 : 3$.

The optimality baseline as shown in Figure 4 is solved by Gurobi 11.0.2 (gur). In order to reproduce the optimality baseline, a license of Gurobi is required. Otherwise, we can resort to open-source operations research libraries such as Google-OR tools (goo). We remark that solving the MILP in Appendix G is computationally expensive, such that it may only apply to small scale datasets such as German Credit. If one wants to compute the optimality baseline for other datasets, then the number of used features needs to be reduced.

The hyperparameters of the models used in the experiment are specified as follows. The models Bagging, GP, RBF, RndForest, AdaBoost, GradBoost, and QDA are scikit-learn (skl) models, where all hyper-parameters are kept default. The models DNN, SVM, RBF, and LR are implemented by PyTorch (Paszke et al., 2019). The DNN has three layers. The SVM uses the linear kernel. The models XGBoost (Chen & Guestrin, 2016) and LightGBM (Ke et al., 2017) are used by their scikit-learn interface, with all hyper-parameters kept default.

## J  POTENTIAL IMPACT

Our proposed framework has several potential positive impacts. By enhancing the transparency and interpretability of machine learning models, our method can improve trust in AI systems, particularly in high-stakes applications such as healthcare, finance, and criminal justice. The ability to generate more concise and actionably efficient counterfactual explanations can aid in identifying and mitigating biases, leading to fairer decision-making processes. Additionally, the framework's versatility and scalability across various models and datasets can democratize access to advanced AI interpretability tools, fostering greater inclusivity in AI development and deployment.

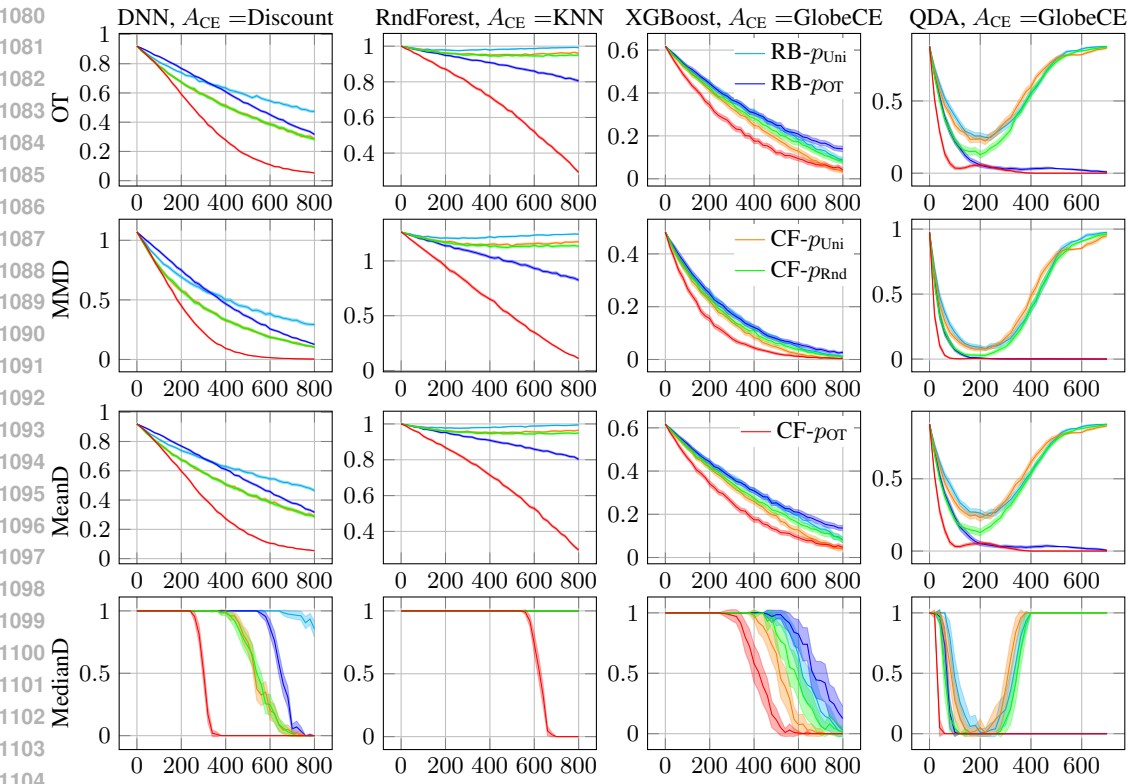

Figure 5: [*HELOC*] $D(f(\mathbf{z}), \mathbf{y}^*)$ vs. allowed actions $C$. Experiments are with 100 runs. The shadows show the 99.9% confidence intervals. The legends apply to all plots. $A_{\text{Value}}^{\text{avg}}$ is used.

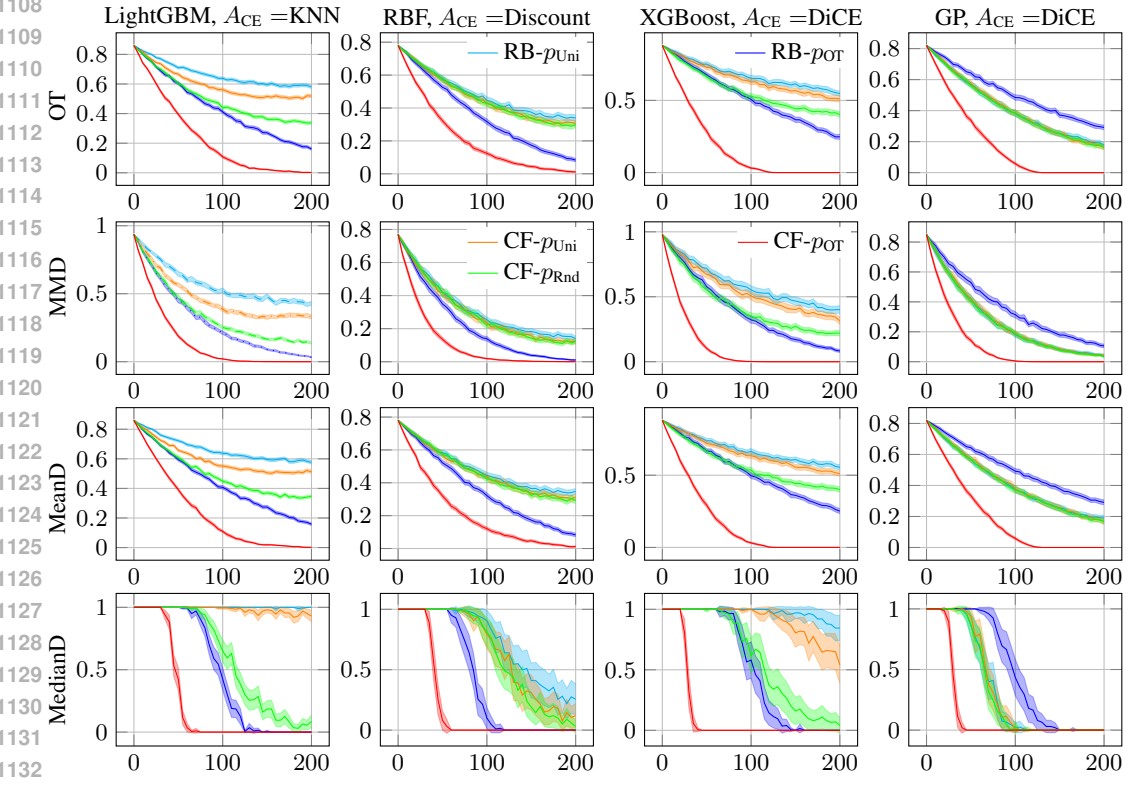

Figure 6: [*German Credit*] $D(f(\mathbf{z}), \mathbf{y}^*)$ vs. allowed actions $C$. Experiments are with 100 runs. The shadows show the 99.9% confidence intervals. The legends apply to all plots. $A_{\text{Value}}^{\text{avg}}$ is used.

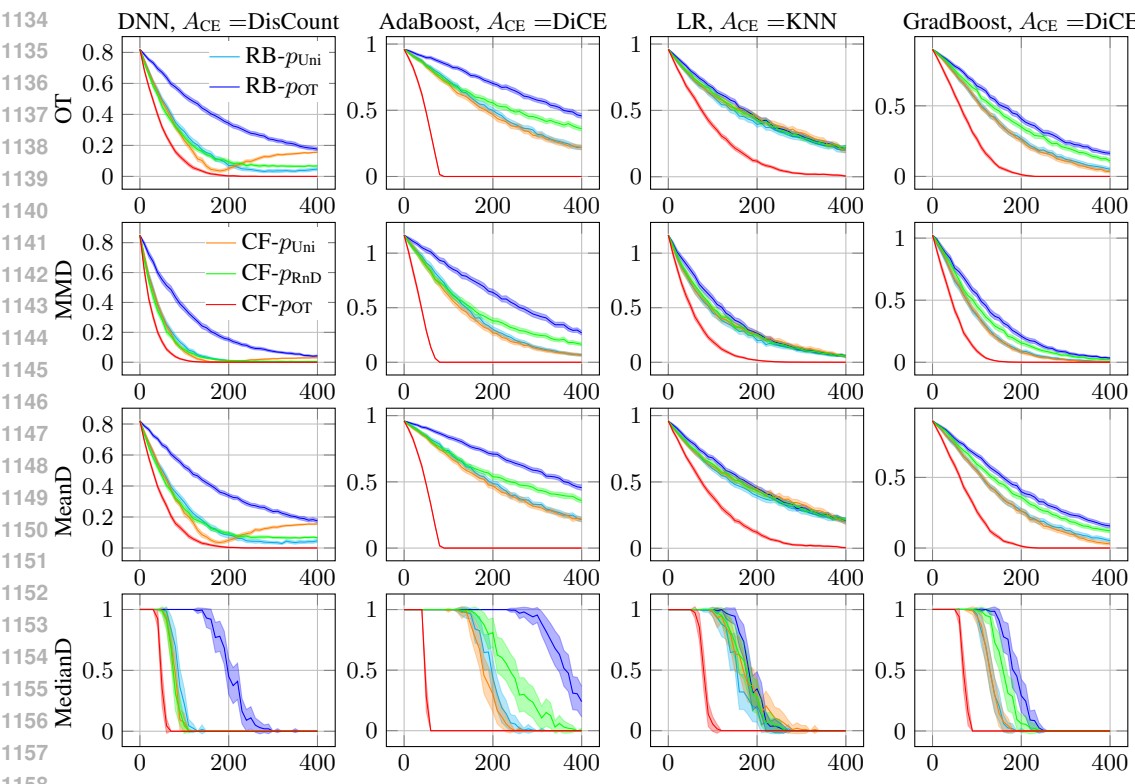

Figure 7: [*Hotel Bookings*] $D(f(\mathbf{z}), \mathbf{y}^*)$ vs. allowed actions $C$. Experiments are with 100 runs. The shadows show the 99.9% confidence intervals. The legends apply to all plots. $A_{\text{Value}}^{\max}$ is used.

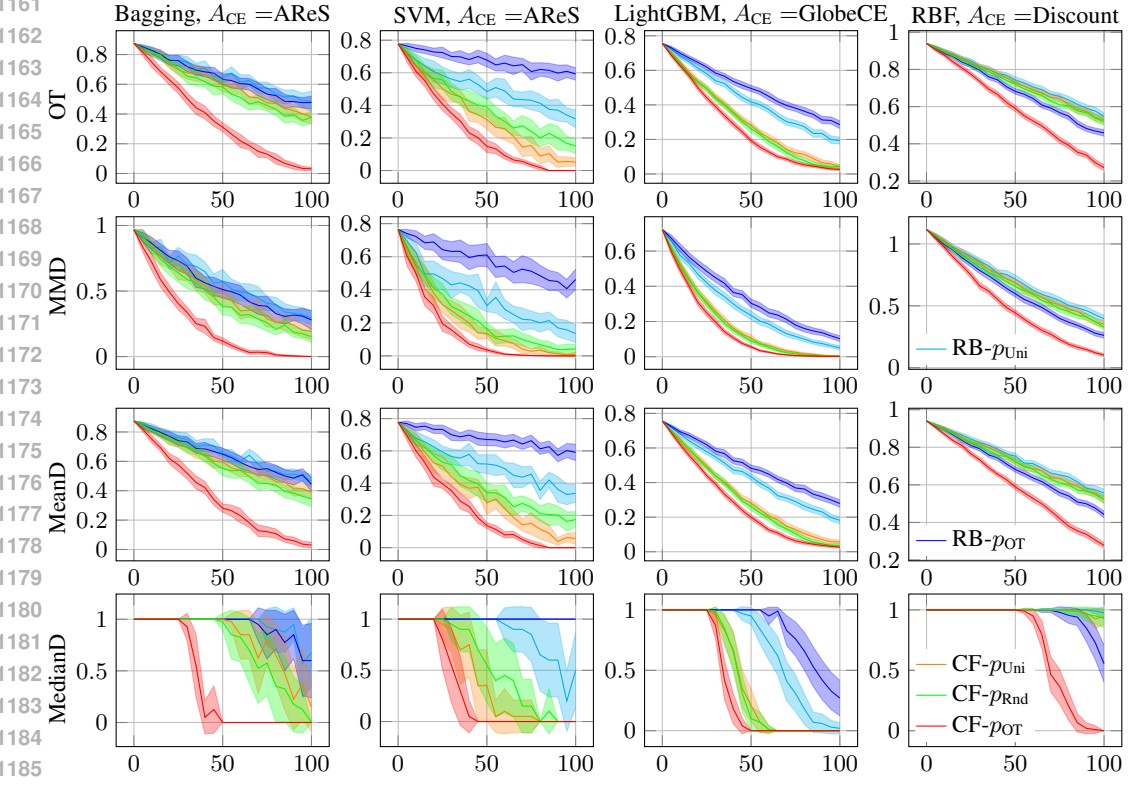

Figure 8: [*COMPAS*] $D(f(\mathbf{z}), \mathbf{y}^*)$ vs. allowed actions $C$. Experiments are with 100 runs. The shadows show the 99.9% confidence intervals. The legends apply to all plots. $A_{\text{Value}}^{\max}$ is used.

