# OpenReview forum: "Refining Counterfactual Explanations With Joint-Distribution-Informed Shapley Towards Actionable Minimality"
_ICLR.cc/2025/Conference — Submitted to ICLR 2025_

### Official Review · Reviewer_WJ7H · 2024-10-30

**Soundness:** 3
**Presentation:** 3
**Contribution:** 3
**Rating:** 8
**Confidence:** 3

**Summary:**

This paper addresses the challenge of creating valid and actionable counterfactual explanations by minimizing unnecessary feature changes. The authors introduce the COLA framework, which computes a joint distribution between observed and counterfactual data to inform Shapley values for feature attributions. This joint-distribution-informed approach allows them to refine the counterfactual explanations without imposing restrictions on model types or algorithms to compute the counterfactuals.

**Strengths:**

This paper addresses the important task of making counterfactual explanations actionable by minimizing unnecessary feature changes.

The authors offer a detailed theoretical basis for their methodology with clear explanations and motivations.

By sharing their code and giving computational details, the author makes the approach more accessible and transparent.

The results seem robust and the counterintuitive finding that relying solely on exact alignment between factual and counterfactual data can be suboptimal is interesting.

**Weaknesses:**

The approach is computationally demanding, mainly due to the optimal transport (OT) calculation, which may limit its scalability in large datasets. (which the authors acknowledge.)

While OT provides a probabilistically informed alignment, it may only sometimes yield the optimal feature attributions for all tasks, as it lacks task-specific tuning and causal considerations. However, the authors denote this as future work.

The approach is primarily evaluated on binary classification and tabular data, leaving its effectiveness on multiclass or regression and other data types (e.g., images, text) scenarios unexplored.

**Questions:**

How well would this method generalize to different multiclass and regression settings?

---

> ### Author Response · Authors · 2024-11-14
>
> We sincerely appreciate the reviewer for recognizing the importance of the problem the paper addresses, the effectiveness of our method, and the robustness of our empirical results.
>
> Responses to the reviewer are as follows. (**Q** and **A** are questions raised by the reviewer and answers provided by the authors, respectively).
>
> **Q1:** “The approach is computationally demanding, mainly due to the optimal transport (OT) calculation, which may limit its scalability in large datasets. (which the authors acknowledge.)”
>
> **A1:** In fact, OT is computed really fast: To compute OT without the entropic term (i.e. $\varepsilon=0$ in (8)), one just solves a linear programming problem fairly fast. To compute OT with the entropic term (i.e. $\varepsilon>0$ in (8)), the convergence is significantly faster, using  Sinkhorn–Knopp algorithm [1].
>
> The main computational cost, in practice, stems from the Shapley value calculation. Our proposed CF-$p_{\text{OT}}$ is as efficient as all other baselines, including RB-$p_{\text{Uni}}$ [2], RB-$p_{\text{OT}}$ (a variant of [2]), CF-$p_{\text{Uni}}$, CF-$p_{\text{Rnd}}$ (variants of [3]), and CF-$p_{\text{Ect}}$ [2,3]. The other operations in COLA (i.e., lines 4-16) are polynomial in both the number of data points and data dimensions, requiring significantly less time than computing the Shapley values.
>
> **Q2:** “While OT provides a probabilistically informed alignment, it may only sometimes yield the optimal feature attributions for all tasks, as it lacks task-specific tuning and causal considerations. However, the authors denote this as future work.”
>
> **A2:** The proposed COLA is a versatile algorithmic framework that can refine explanations generated by other task-specific CE algorithms. As the reviewer suggested, exploring how to tune COLA towards further adapting to a specific task and considering causal graphs (assumed available) is worth exploring. Thank you for pointing out.
>
> **Q3:** “The approach is primarily evaluated on binary classification and tabular data, leaving its effectiveness on multiclass or regression and other data types (e.g., images, text) scenarios unexplored. How well would this method generalize to different multiclass and regression settings?”
>
> **A3:** Though our experiments are done for binary classification tasks, we remark that this paper's proposed **COLA and $p$-SHAP apply directly to multiclass classification, regression, and image classifications**. We refer to the illustration in Figure 2 to explain this, as follows.
> - For multiclass classification, lines 1-2 are unaffected. Lines 3-4 involve the computation of the Shapley value, which functions for both binary and multiclass classification tasks (similar to the SHAP Python library), so these steps also remain unchanged. Line 5 and lines 6-16 involve matrix operations on the data, which likewise stay the same.
> - For regression, the only difference compared to binary classification is that the desired counterfactual effect (or target model output), $\mathbf{y}^*$, is a real value. After determining $\mathbf{y}^*$ in line 1 of COLA, the rest of the process is independent of $\mathbf{y}^*$, meaning that all subsequent steps remain unchanged.
> - For (binary or multiclass) image classification, the pixels of an image can be treated as a vector. Specifically, in lines 1-2 of Figure 2, each row in $\mathbf{x}$ and $\mathbf{r}$ would represent an image. Thus, all steps remain the same.
>
> Once again, we thank the reviewer for their insightful comments, and we are open to any further discussions or questions.
>
> ### References
> [1] Knight, Philip A. "The Sinkhorn–Knopp algorithm: convergence and applications." SIAM Journal on Matrix Analysis and Applications 30.1 (2008): 261-275.
>
> [2] Scott M Lundberg and Su-In Lee. A unified approach to interpreting model predictions. Advances in neural information processing systems, 30, 2017.
>
> [3] Albini, Emanuele, et al. "Counterfactual shapley additive explanations." Proceedings of the 2022 ACM Conference on Fairness, Accountability, and Transparency

---

> > ### Comment · Reviewer_WJ7H · 2024-11-24
> >
> > Thank you for the additional clarifications, especially A3.

---

### Official Review · Reviewer_hZrx · 2024-11-03

**Soundness:** 3
**Presentation:** 3
**Contribution:** 1
**Rating:** 6
**Confidence:** 5

**Summary:**

The paper proposes an algorithm to compute CFEs for datapoints such that a minimal number of features are changed, thereby generating sparse CFEs.

**Strengths:**

The paper establishes an excellent mathematical framework to generate CFEs such that they change the smallest number of features from the original datapoint. This framework connects CFEs with Feature attribution technique SHAP.

**Weaknesses:**

There are several weakness in the paper, I list them as follows. The authors only need to address the ones marked as major for the rebuttal (if they like to):
1. Line 51: "some CE algorithms assume differentiable models, whereas others are designed specifically for tree-based or ensemble models" -- this is misleading, as many CE algorithms work for black-box ML models, you can find them in the Verma et al. survey paper.

2. [Major] Line 79: "Three major challenges remain in addressing this problem". The next three lines do not sounds like challenges at all. The first one says that one CE algorithm will not suffice (which is not true as past work have proposed algorithms that work for generating sparse CEs), the second one about not assuming anything about the model (this is not novel several works have done this), and the third is about how feature attribution techniques can be misleading (which is obvious because they are not optimized to be used as CE, a feature that is marked as important by some FA method does not necessarily mean that it is an effective way to change the prediction). Please state the challenges (if there are any).

3. Line 124 $M_{c_{ik}}$ is not defined which makes understanding Eq 1(d) and 1(e) hard. Also how does having $c_{ik}$ = 0 allows no change in its value? I don't see that being implied from the equation

4. B-SHAP and RB-SHAP have different background distributions and it is clear to the community about the advantages and disadvantages of each. You define CF-SHAP, but never state what trade-off does one expect when using background distribution from CF distribution instead of the training distribution. Please explain what should one expect from such a FA method.

5. Line 197, correct the typo from RB-SHAP to B-SHAP.

6. The first paragraph of section 4 is obvious, it is just a generalization of the notations from Equations 4-6, I don't think there is a need to give so much attention and explanation for it.

7. [Really Major]. You mention a lot of details about the optimal transport thing to compute the most sparse CE, and do a really good job at explaining Algorithm 1 in Section 5. At this point, I am excited and was anticipating some great experimental results because you spent such a large part of the paper establishing strong mathematical grounds for generating CFEs and even proved some theorems, but the experimental section is **extremely weak** and does not convince me at all about the relevance of the propose technique.
       1. First and the most important weakness is that you need another CFE generation technique to generate a CFEs for a lot of datapoints that you use to compute the joint distribution and then generate a CFE using your technique, is that right? If that is the case, in the paragraph of computation complexity, why do you not include that? Also it is not the case that your technique just takes a CFE from some other technique and refines it, it needs a whole lot of datapoints and CFEs to even start functioning. Please correct me if I am wrong.

       2. The experimental results are very weak for the following reasons:
              **a.)** Result 1 in Section 6 (lines 415 - 421) are not useful when a datapoint does not achieve the desired classification. In other words, if a datapoint does not achieve y* classification (which makes it a valid CFE), there is no point in mentioning the sparsity of that, because it is an invalid and not useful CFE. Therefore the third column in Table 3 is not useful at all.

             **b.)** Several baselines are mentioned in lines 368-372, but none of them are used in the experimental results section? Why? Aren't these the baselines that you need to compare against to show that your technique is better than them atleast in the one metric you are targeting (which is sparsity)? Instead what you do in Table 3 is compare against 5 variations of your own proposed method. Sure one of your own method is better than the other 4 you mention, that does not tell me anything about how does COLA compare to previous techniques. Please let me know if you disagree. Therefore Result 2 is not useful.

             **c.)** The baselines considered in the paper (the results for which are not reported atleast in the main paper), are outdated and several new techniques have been proposed that are better at all metrics than DiCE and KNN, for e.g. see Amortized Generation of Sequential Algorithmic Recourses for Black-box Models.

            **d.)** An underlying and unstated key assumption in the formulation of the paper is that sparsity is the most important metric to consider when generating CFEs. This is atleast not agreed upon by the community, see the list of metrics proposed in Verma et al. survey paper.

**Questions:**

I have stated all the questions in the weakness section of the review. The paper needs to really strengthen its experimental results section. I would expect something like a table, where the columns are the various metrics along with a CFE is evaluated, like validity, proximity, sparsity, distance of training data manifold, adherence to causal constraints, and the time to generate the CFEs (including dependence on any other technique) and the rows to be a large set of baselines. Only when the numerical results demonstrate the superiority of COLA, can the supporting theory, theorems, and the claims of near-optimal performance be useful.

---

> ### Author Response · Authors · 2024-11-13
>
> ### Clarification of Misunderstandings
>
> We would like to point out some misunderstandings of Reviewer hZrx and clarify several important facts before replying to their specific comments:
>
> **[Fact 1]**: The paper does NOT propose a new CE method. Instead, there are two proposed things: One is the proposed OT-based p-SHAP, used for feature attributions for refining counterfactual explanations. The other is a versatile mechanism to refine/sparsify explanations generated by other CE methods. We have clearly stated this in the main contributions of Section 1.
>
> **[Fact 2]**: To our knowledge, this is the first work that provides a unified way for refining/sparsifying explanations generated by existing tabular data CE methods, without assumptions on the machine learning models and the CE methods. We’d happily refer to others’ work for this if there is any.
>
> **[Fact 3]**: Due to [Fact 1], the baselines in this paper are not CE methods.
> 1) To demonstrate the validity and effectiveness of COLA, we have provided its empirical results (in Figures 3 and 4, and Appendix H) under different scenarios (i.e. using COLA to refine the generated counterfactuals from different CE methods for different ML algorithms).
>
> 2) To explore the potential and importance of using joint distribution informed by Shapley to refine counterfactuals and demonstrate the proposed p-SHAP, we have provided its empirical results in Table 3.  Our baselines are those different Shapley methods and different joint probability distribution setups. In Table 2, CF-$p_{\text{OT}}$ stands for our proposed OT-based p-SHAP. All the other rows are baselines. We benchmarked our proposed p-SHAP to all the baselines in all the mentioned CE methods stated in lines 368-372.
>
> ### Response to "7.[Really Major]"
>
> We disagree with the reviewer about the weakness of the experiment. We have benchmarked our proposed p-SHAP and some other existing Shapley methods (baselines) on 4 datasets, under different scenarios (changing in 5 different CE algorithms and 13 different machine learning algorithms). We believe the experimental results strongly indicated the effectiveness of our proposed COLA. Besides, **the strength of our experimental results has been explicitly acknowledged by Reviewers iyvf, hbCQ, and WJ7H** regarding its exhaustiveness and robustness.
>
> More detailed responses to the reviewer are as follows. (**Q** and **A** are question raised by the reviewer and answer provided by the authors respectively).
>
> **Q1**: “1. First and the most important weakness is that you need another CFE generation technique to generate a CFEs for a lot of datapoints that you use to compute the joint distribution and then generate a CFE using your technique, is that right? If that is the case, in the paragraph of computation complexity, why do you not include that? Also it is not the case that your technique just takes a CFE from some other technique and refines it, it needs a whole lot of datapoints and CFEs to even start functioning. Please correct me if I am wrong.”
>
> **A1**: **What you said is not correct** for three reasons.
>
> First, the proposed COLA is not a CE (referred to by the reviewer as “CFE”) algorithm. Instead, it is a unified mechanism to refine (or sparsify) explanations generated by any CE algorithm (See [Fact 1]). It is a sparsification mechanism for other CE methods, yet does NOT rely on one specific CE method.
>
> Second, we have already considered the computation complexity of other CE methods when analyzing the computational complexity. See Lines 348-353 “Complexity of COLA”. Note that $A_{\text{CE}}$ is the CE algorithm in step 1, and  $O(M_{\text{CE}})$ indicates its complexity.
>
> Third, our proposed COLA does NOT need many data points and CEs to work. In fact, it also work with one single factual data point and one corresponding CE data point. This is exactly the case of CF-$p_{\text{Ect}}$ stated in the last row of Table 2, where each factual data point is known to be aligned with another exact counterfactual data point. As we can see from the corresponding results in Figure 4, COLA functions well in this case. The concrete number of the used data points depends on the used CE method.

---

> ### Author Response · Authors · 2024-11-13
>
> **Q2**: “2. The experimental results are very weak for the following reasons: a.) Result 1 in Section 6 (lines 415 - 421) are not useful when a datapoint does not achieve the desired classification. In other words, if a datapoint does not achieve y* classification (which makes it a valid CFE), there is no point in mentioning the sparsity of that, because it is an invalid and not useful CFE. Therefore the third column in Table 3 is not useful at all. ”
>
> **A2**:  The reviewer **misunderstood** the mechanism of COLA.
>
> As COLA is to refine/sparsify the explanations generated by other CE methods (see [Fact 1]), that is to say, the to-be-refined explanations already achieve the desired classification (referred to as “the target outcome $\mathbf{y}^*$” in line 269, which is formally defined as $\mathbf{y}^*=f(\mathbf{r})$ in line 260). Otherwise, one would seek an alternative CE method rather than the selected one to obtain the desired classification first. Therefore, The third and fourth columns in Table 3 suggest exactly how good sparsity our method COLA could achieve compared to not using it.
>
> Besides, to our best knowledge, none of the hundreds of CE methods surveyed in [1] could guarantee an arbitrary desired counterfactual effect. We’d appreciate it if the reviewer would clarify the question further in case it bothers you.
>
> **Q3**: “b.) Several baselines are mentioned in lines 368-372, but none of them are used in the experimental results section? Why? Aren't these the baselines that you need to compare against to show that your technique is better than them atleast in the one metric you are targeting (which is sparsity)? Instead what you do in Table 3 is compare against 5 variations of your own proposed method. Sure one of your own method is better than the other 4 you mention, that does not tell me anything about how does COLA compare to previous techniques. Please let me know if you disagree. Therefore Result 2 is not useful. ”
>
> **A3**:  **These stay entirely away from the truth**.
>
> We clarify in [Fact 3] that: 1) lines 368-372 indicate the CE algorithms (not baselines) that we used in the proposed COLA, to evaluate its versatility, and the empirical results are provided in Figures 3 and 4 and Appendix H; 2) the baselines to evaluate the proposed p-SHAP are set in Table 2 and empirical results are provided in Table 3, Figures 3 and 4, and appendix H; 3) other rows in Table 3 except CF_$p_\text{OT}$ are not 5 variations of our own proposed methods.
>
> In fact, RB-$p_{\text{Uni}}$ is exactly the random Shapley proposed in [2]. RB-$p_{\text{Uni}}$ follows the random Shapley in [2] but changes the joint distribution to the same as CF-$p_{\text{ot}}$ for the ablation study. CF-$p_{\text{Uni}}$ and CF-$p_{\text{Rnd}}$ are the Counterfactual Shapley proposed in [3]. Since it assumes a known counterfactual distribution conditional on each factual data point, whereas it is unknown in our experiment, we used $p_{\text{Uni}}$ and $ p_{\text{Rnd}}$ instead. CF-$p_{\text{Ect}}$ is exactly the Counterfactual Shapley proposed in [3]. When there is a one-to-one alignment between factual and counterfactual data points, Counterfactual Shapley degrades to the Baseline Shapley proposed in [2].
>
> Hence, **it is NOT true that we compare our proposed one with its variations**.
>
>  **Q4:** “c.) The baselines considered in the paper (the results for which are not reported atleast in the main paper), are outdated and several new techniques have been proposed that are better at all metrics than DiCE and KNN, for e.g. see Amortized Generation of Sequential Algorithmic Recourses for Black-box Models.”
>
> **A4**:  **This is simply not true**.
>
> As we mention before, CE algorithms are not baselines, and all their results have been reported in Table 3, Figures 3 and 4, as well as Appendix H. Besides, **these CE methods that we use are not outdated, either**. In fact, DiCE [4] was proposed in 2020 and is still the most commonly used CE method in the industry and academia nowadays; KNN (as a CE method) [3] was proposed in 2022; AReS [5] was proposed in 2020; GLOBE [6] was proposed in 2023, and Discount emerged in early 2024. By contrast, the specific literature [7] mentioned by the reviewer emerged in 2021 on arXiv and published in early 2022. Moreover, **we do not see why the performance of CE methods matters here**, because the proposed COLA can refine explanations generated by any CE methods (including [7]) regardless of which performs better than the others.

---

> ### Author Response · Authors · 2024-11-13
>
> **Q5:** “d.) An underlying and unstated key assumption in the formulation of the paper is that sparsity is the most important metric to consider when generating CFEs. This is atleast not agreed upon by the community, see the list of metrics proposed in Verma et al. survey paper.”
>
> **A5**:  **We never had such assumption at all**.
>
> Our goal of seeking more sparse explanations is **NOT** equivalent to “we assuming that sparsity is the most important metric to consider when generating CE explanations”. It is out of the scope of this paper to discuss which metric is the most important to be considered when generating CE explanations.
>
> In our opinion, which metric is the most important would-be task-specific. We refer to [Fact 2], that one should select a proper CE method, and, if sparsity is desired, then apply our proposed method to refine the explanations.
>
> **Q6:** “The paper needs to really strengthen its experimental results section. I would expect something like a table, where the columns are the various metrics along with a CFE is evaluated, like validity, proximity, sparsity, distance of training data manifold, adherence to causal constraints, and the time to generate the CFEs (including dependence on any other technique) and the rows to be a large set of baselines.Only when the numerical results demonstrate the superiority of COLA, can the supporting theory, theorems, and the claims of near-optimal performance be useful.”
>
> **A6**:  **We have already reported validity, proximity, and sparsity** in Table 3. Namely,
> - Validity is referred to as “counterfactual effect” and is shown in columns 3 and 4 in Table 3, and we are showing how the refined explanations remain the desired counterfactual with significantly better sparsity.
> - Proximity is $\|\mathbf{z}-\mathbf{x}\|$, and we already reported in the subcolumns of columns 3 and 4 in Table 3 (normalized by $\|\mathbf{r}-\mathbf{x}\|$).
> - Sparsity is reported as the “# of Modified Features” in columns 3 and 4. Our proposed Shapley method is benchmarked towards other four Shapley methods proposed in [2] and [3] combined with different joint probability distributions setups.
>
> Our experiments are designed to demonstrate the versatility of the proposed COLA and the effectiveness of the proposed p-SHAP. (See A3 above).
>
> At last, as explicitly mentioned by the other **Reviewers iyvf, hbcQ, WJ7H**, our experiments are **exhaustive and robust**. Thus, we argue that **this judgment of “extremely weak experiments” is not fair**.
>
> ### Response to “2. [Major]”
>
> The problem being addressed in this paper is challenging because:
>
> - (For the first challenge mentioned) To our best knowledge, existing sparse CE algorithms are task-specific, which means none of them can be used solely for various tasks. It’s unrealistic that expecting one single existing (sparse) CE algorithm would meet all the needs universally for various CE tasks.
> Also, except for this paper, there exist no refinement mechanism that works for arbitrary models or CE tasks (see [Fact 2]).
> Note that the proposed COLA is a sparsification mechanism used to refine the explanations generated from existing CE algorithms. So even if there were already some “universally applicable sparse CE method”, the proposed COLA would still be applicable to it in order to sparsify its generated explanations further (see [Fact 1]).
> Hence, **whether a general-purpose sparsification mechanism is possible remains open and challenging**.
> - (For the second challenge mentioned) First, existing *sparse* CE algorithms usually have some assumptions about the model (e.g., model differentiability, model architecture, available causal graph, specific tasks, focused CE metrics, etc.). If the reviewer could point out a concrete work, we’d happily refer to the work in our paper (see [Fact 2]). Second, the question of **whether a versatile mechanism exists without assumptions about the model remained open and challenging** before our work in this paper.
> - (For the third challenge mentioned). Some researchers have tried to use FA in finding CEs [3]. However, it is *obviously* challenging to integrate FA (like Shapley) to improve CE, because FA mechanisms are not specifically or essentially designed for CE.  Using FA like feature importance to find CEs may be misleading because, as the reviewer stated, “A feature that is marked as important by some FA method does not necessarily mean that it is an effective way to change the prediction”.
> So **finding an effective FA that also works for CE tasks is challenging**, which is addressed in this paper. It’s also worth noting that our proposed method p-SHAP is demonstrated to outperform existing work [2,3] in finding proper FAs for CEs and obtaining sparser CEs.

---

> ### Author Response · Authors · 2024-11-13
>
> ### Response to Other "Minor Points 1,3,4,5,6"
>
> **Q7** “1. Line 51: "some CE algorithms assume differentiable models, whereas others are designed specifically for tree-based or ensemble models" -- this is misleading, as many CE algorithms work for black-box ML models, you can find them in the Verma et al. survey paper.”
>
> **A7:**  We would like to **take the reviewer’s suggestion** and revise the sentence to be “To our best knowledge, there has not been an unified way to refine counterfactuals generated by arbitrary CE algorithms without assumptions on models.”
>
> **Q8:** “3. Line 124 $M_{c_{ik}}$ is not defined which makes understanding Eq 1(d) and 1(e) hard. Also how does having $c_{ik} = 0$ allows no change in its value? I don't see that being implied from the equation”
>
> **A8:**  **There is no $M_{c_{ik}}$ defined in our paper**. In the following response, we assume the reviewer was referring to the notation $M$ that is multiplied to the variable $c_{ik}$. This is a technique in writing mathematical formulation, and is stated in line 134: “….if $c_{ik}=1$, remark that $M$ is a sufficiently large constant such that $z_{ik}$ has good freedom to change.” To further explain it, if $c_{ik}=1$, (1d) and (1e) becomes $-M\leq z_{ik}\leq M$. Since $M$ is large enough, $z_{ik}$ has sufficient freedom. On the contrary, if $c_{ik}=0$, (1d) and (1e) become $z_{ik}\leq x_{ik}$ and $z_{ik}\geq x_{ik}$ respectively, and they have to be satisfied simultaneously. Hence $z_{ik}=x_{ik}$, which means that for any $i$ and $k$, $z_{ik}$ is not allowed to change to any value other than $x_{ik}$.
>
> **Q9:** “4. B-SHAP and RB-SHAP have different background distributions and it is clear to the community about the advantages and disadvantages of each. You define CF-SHAP, but never state what trade-off does one expect when using background distribution from CF distribution instead of the training distribution. Please explain what should one expect from such a FA method.”
>
> **A9:**  **This is simply not true**. It is not us who defined CF-SHAP. The researchers propose the CF-SHAP method in [3] and we just cited their work as shown in lines 186-187. As indicated by the authors in [3], CF-SHAP outperforms the Shapley methods that do not use CF distribution when one wants to do contrastive feature attribution. This conclusion aligns exactly with our results in the paper. However, remark that the distribution of counterfactuals is not always available, imposing limitations on this method. Our results further demonstrate that even if such a counterfactual distribution is known, it may not be the best option, as the proposed p-SHAP can outperform it.
>
> **Q10:** “5. Line 197, correct the typo from RB-SHAP to B-SHAP.”
>
> **A10:**  **Thank you for this**. We will correct the typo in our revised version.
>
> **Q11:** “6. The first paragraph of section 4 is obvious, it is just a generalization of the notations from Equations 4-6, I don't think there is a need to give so much attention and explanation for it.”
>
> **A11:**  **Disagree.** Ours in (7a) and (7b) differ fundamentally with (4)-(6), explained as follows. The CF-SHAP proposed in [3] replies on a concrete counterfactual algorithm to define the counterfactual distribution $\mathcal{D}$ in (6). Hence, CF-SHAP is dependent on a CE algorithm. Our proposed p-SHAP, however, is completely independent of CE algorithms. As for (4) and (5), they do not use counterfactual distribution and differ from ours. This is why this paragraph is mandatory. And we would like to extend this part further in the revised version.
>
> ### References
> [1] Guidotti, Riccardo. "Counterfactual explanations and how to find them: literature review and benchmarking." Data Mining and Knowledge Discovery 38.5 (2024): 2770-2824.
>
> [2] Scott M Lundberg and Su-In Lee. A unified approach to interpreting model predictions. Advances in neural information processing systems, 30, 2017.
>
> [3] Albini, Emanuele, et al. "Counterfactual shapley additive explanations." Proceedings of the 2022 ACM Conference on Fairness, Accountability, and Transparency. 2022.
>
> [4] Ramaravind K. Mothilal, Amit Sharma, and Chenhao Tan (2020). Explaining machine learning classifiers through diverse counterfactual explanations. Proceedings of the 2020 Conference on Fairness, Accountability, and Transparency.
>
> [5] Kaivalya Rawal and Himabindu Lakkaraju. Beyond individualized recourse: Interpretable and interactive summaries of actionable recourses. Advances in Neural Information Processing Systems, 33:12187–12198, 2020.
>
> [6] Dan Ley, Saumitra Mishra, and Daniele Magazzeni. Globe-ce: a translation based approach for global counterfactual explanations. In International Conference on Machine Learning, pp. 19315–19342. PMLR, 2023.
>
> [7] Verma, Sahil, Keegan Hines, and John P. Dickerson. "Amortized generation of sequential algorithmic recourses for black-box models." Proceedings of the AAAI Conference on Artificial Intelligence. Vol. 36. No. 8. 2022.

---

> > ### Comment · Reviewer_hbCQ · 2024-11-25
> > **The objective of COLA**
> >
> > While this thread has many important considerations, I would like to emphasise and agree with the authors about the objective of COLA, proposed in the paper. As I understand it, this not a new CE method, but a technique that refines a set of existing counterfactual explanations, which may be obtained in any way.
> >
> > This misunderstanding seems to nullify much (but not all) of the later comments, and it would be good if we can all come to a common understanding.
> >
> > Currently, I am inclined to think that the author rebuttal about the objective of COLA is correct. The reviewer states they are very confident of their assessment (more so than I am, at least). Perhaps it would be good for them to clarify further about why they feel COLA is a CE method, because if it isn't, then a lot of this feedback would be rendered inapplicable.

---

> > > ### Comment · Reviewer_hZrx · 2024-11-28
> > > **Reviewer Response**
> > >
> > > I thank the authors for their time and effort in understanding the paper. After reading the rebuttal, it definitely seems like I misunderstood the paper and its contributions, and the rebuttal has resolved most of my concerns.
> > >
> > > I have updated the score in the light of the rebuttal. Thank you again!

---

> > > > ### Author Response · Authors · 2024-11-28
> > > > **Thank you!**
> > > >
> > > > We thank the reviewer hZrx for the efforts on re-evaluating our work! We are delighted to know that the concerns have been resolved!

---

### Official Review · Reviewer_hbCQ · 2024-11-03

**Soundness:** 3
**Presentation:** 2
**Contribution:** 3
**Rating:** 6
**Confidence:** 3

**Summary:**

The paper presents a post-hoc modification of a set of counterfactual explanations. It does so with the aim of reducing the modification-costs of the counterfactual explanations found. To do so, it employs a new variant of the common SHAP feature attribution technique, which explicitly includes both in-distribution and counterfactual-distribution data in the SHAP baseline prediction computation. The feature attributions so obtained are used to modify the existing set of counterfactual explanations, and it turns out that this modification reduces the costs associated with the counterfactuals.

**Strengths:**

The paper presents a new problem perviously unreported in the literature, and solves it effectively. The empirical findings seem to be robust and diverse. The paper presents the proposed method adequately and displays the improvements it provides through exhaustive experiments.

**Weaknesses:**

The writing could be clearer. The paper introduces and repeatedly uses terms without defining them: "exact alignment" (line 25, 197, etc; finally explained only on line 308), "well-performed action plans" (line 98), etc. It would be useful to define the terms at the time of their first use.

I do not understand the "decoupling" phenomenon mentioned in line 85-86, and the reference to the later demonstration was lost (or I may have missed it). Perhaps more exposition and an explicit reference to where in the paper to find this "later" would be helpful.

The "counter-intuitive finding" described in lines 101-105 remained difficult to parse until I had reached the end of the paper, because "associating" and "alignment" had not been defined yet. These terms can be defined and explained earlier in the paper, so that the findings become easier to understand in a first read. The abstract could also be amended similarly to ensure that it is easier to understand.

It is unclear to me why the proposed SHAP variant performs better than other SHAP variants, or indeed why would SHAP feature attributions be a good way to perform post-hoc optimisations on generated counterfactual explanations at all. The paper presents the proposed method adequately and displays the improvements it confers, but I'm unsure why this is the case. Any intuition or ideas that the authors have about this would be good to include. While additional experiments aren't strictly necessary, I am curious about how non SHAP based methods would perform. I have not looked at the code directly, but if the results are easily replicable, this might lead to rapid future work in this area, providing more understanding of the phenomena reported in the paper.

**Questions:**

Reordering the writing would help readers make sense of the contributions in a single pass. The inclusion of SHAP seems arbitrary - and although demonstrated to work well - it is unclear why other methods are not included.

---

> ### Author Response · Authors · 2024-11-14
>
> We thank the reviewer for recognizing the novelty and effectiveness of our proposed method, and we greatly appreciate the acknowledgment of our exhaustive experiment results.
>
> Responses to the reviewer are as follows. (**Q** and **A** are questions raised by the reviewer and answers provided by the authors, respectively).
>
> ### Intuition Behind $p$-SHAP and Why It Performs Well
>
> We first address the reviewer's concern mentioned in the final point of weaknesses and reiterated in the questions: specifically, the intuition behind why p-SHAP performs better than other variants, as discussed in Q1 and A1 below.
>
> **Q1:** “It is unclear to me why the proposed SHAP variant performs better than other SHAP variants, or indeed why would SHAP feature attributions be a good way to perform post-hoc optimisations on generated counterfactual explanations at all. The paper presents the proposed method adequately and displays the improvements it confers, but I'm unsure why this is the case. Any intuition or ideas that the authors have about this would be good to include. While additional experiments aren't strictly necessary, I am curious about how non SHAP based methods would perform. I have not looked at the code directly, but if the results are easily replicable, this might lead to rapid future work in this area, providing more understanding of the phenomena reported in the paper. Reordering the writing would help readers make sense of the contributions in a single pass. The inclusion of SHAP seems arbitrary - and although demonstrated to work well - it is unclear why other methods are not included.”
>
> **A1:** Thank you for acknowledging the contributions and impact of our proposed method. We would like to try our best to explain why SHAP feature attribution, if used properly, is a good way to perform post-hoc optimization for CE towards fewer modifications, detailed as follows.
>
> (**Intuition behind using FA**) To create a unified approach for refining generated explanations with minimal changes, a simple idea is to determine which features are most important to modify for each data point to achieve the desired counterfactual outcome. Feature attribution (FA) is for this purpose exactly.
>
> (**Intuition behind using Shapley**) Previous research has shown that performing feature attribution (FA) separately from counterfactual explanations (CE) does not work well for determining the importance of features in achieving a counterfactual outcome [1,2]. Therefore, it makes sense to integrate CE information into the FA process, which should help the attribution results better capture the true importance of each feature in relation to the counterfactual effect. And [1] proposed CF-SHAP, which uses counterfactual data distribution $\mathcal{D}$. It is shown in [1] that CF-SHAP performs better than the other Shapley methods. This result also aligns with ours in Section 6 and Appendix H.
>
> (**An identified issue for using Shapley with CE**) Yet, CF-SHAP assumes a known counterfactual data distribution (determined by a selected CF algorithm; see [1]). We remark that this sounds tricky: The factual data are collected from real-world and hence their distribution is determined by the physical world, staying **objective**. However, the counterfactual data distribution $\mathcal{D}$ need to be generated from a CE algorithm. Remark that there are hundreds of CE algorithms, and selecting which to generate $\mathcal{D}$ is **subjective**. It is important to note that FA is to identify the importance of features of a model’s input. However, the FA method CF-SHAP yields different importance scores to the same input data of a given model: Consider one counterfactual data point $r_0$ generated by two different CE methods “CE-a” and “CE-b”, for an observed factual $x_0$. Since the model, the factual, and the counterfactual are all fixed, the FA results should be determined and unique. But the FA method CF-SHAP would yield different FA results based on which of “CE-a” and “CE-b” is used to generate $\mathcal{D}$. This thought experiment motivates us to re-think the real need of an FA method for CE, and propose something new to utilize the CE information better rather than associating it with $\mathcal{D}$ in Shapley value computation.

---

> ### Author Response · Authors · 2024-11-14
>
> (**Intuition of why $p$-SHAP performs better**) Let us revisit our core objective: First, we want to minimize the dissimilarity of the factual $\mathbf{x}$ and a to-be-found counterfactual $\mathbf{z}$ to achieve the counterfactual effect $\mathbf{y}^*$. Second, we want $\mathbf{z}$ is more similar than $\mathbf{x}$ compared to the original counterfactual $\mathbf{r}$. We want an FA method that satisfies both the two needs above. The proposed p-SHAP does so with support from theoretical aspects: The first need is supported by Theorem 4.1, namely, optimal transport (OT) based $p$-SHAP does attribution towards minimizing a upper bound of the dissimilarity between the counterfactual classification $f(\mathbf{z})$ and the desired counterfactual effect $\mathbf{y}^*$. The second need is supported by Theorem 5.1, namely, OT-based $p$-SHAP, used in our proposed COLA, achieves better proximity than the original counterfactual under the Frobenius norm. Importantly, the joint distribution leverages CE information to guide $p$-SHAP for effective feature attribution, while $p$-SHAP itself remains independent of the specific CE algorithm used.
>
> In the revised version, we will slightly reorder the content and provide a brief summary of the intuition outlined above, to help readers better understand why p-SHAP outperforms other methods, as demonstrated by our empirical results.

---

> > ### Author Response · Authors · 2024-11-14
> >
> > ### Responses to Other Points
> >
> > We then reply to the other points raised, as Q2-Q4 and A2-A4 below.
> >
> > **Q2:** “The writing could be clearer. The paper introduces and repeatedly uses terms without defining them: "exact alignment" (line 25, 197, etc; finally explained only on line 308), "well-performed action plans" (line 98), etc. It would be useful to define the terms at the time of their first use.”
> >
> > **A2:** Thank you for your good suggestions! In the revised version, we will modify line 25 to be
> > > … it may be misleading to rely on a counterfactual distribution defined by the CE generation mechanism ….
> > This is because the exact alignment literally refers to a known distribution of a counterfactual, which is usually defined by a given CE method [1].
> > Then, we will modify line 197 to be
> > > …. when $A_{\text{Prob}}$ defines a joint distribution between $\mathbf{x}$ and $\mathbf{r}$, indicating an $i\leftrightarrow j$ alignment of any $\mathbf{x}_{i}$ and $\mathbf{r}_j$.
> >
> > This modification reinforces the concept that the distribution determines the alignment, as stated in line 308.
> >  Besides, we will clarify the term “action plan” in Figure 1 and its caption, in the revised version.
> >
> > **Q3:** “I do not understand the "decoupling" phenomenon mentioned in line 85-86, and the reference to the later demonstration was lost (or I may have missed it). Perhaps more exposition and an explicit reference to where in the paper to find this "later" would be helpful.”
> >
> > **A3:** The original sentence in lines 84-86 reads:
> > > … it is not effective to decouple FA with CE in order to select the most important features to change. We will demonstrate later that this decoupling can result in counterproductive feature modifications …
> > The word “decouple” here, means that FA is performed **independently** of CE. For example, B-SHAP in (4) and RB-SHAP in (5) are defined independently of CE (i.e. it does not use any information from the CE mechanism or the generated CE data points). Such FA lead to counterproductive feature modifications, as shown by Figure 3 with the curves RB-$p_{\text{Uni}}$ and RB-$p_{\text{OT}}$.
> >
> > In the revised version, we will modify line 85 as
> >
> > > ….. it is not effective to perform FA independently of CE to select ….. We will demonstrate later that this decoupling can result in counterproductive feature modifications in the empirical results in Result II of Section 6.
> >
> > **Q4:** “The "counter-intuitive finding" described in lines 101-105 remained difficult to parse until I had reached the end of the paper, because "associating" and "alignment" had not been defined yet. These terms can be defined and explained earlier in the paper, so that the findings become easier to understand in a first read. The abstract could also be amended similarly to ensure that it is easier to understand.”
> >
> > **A4:** We agree with the reviewer that clarifying these terms earlier would be more reader-friendly. The modifications made in A1 above will do the work. Additionally, in line 204, we will further clarify
> > > Contrary to common expectations, we demonstrate that OT can be more effective than relying on a counterfactual distribution defined by the CE generation mechanism done by [1], in Result II of Section 6 later.
> >
> > Once again, we thank the reviewer for their time and effort in helping us improve the manuscript. We remain open to any further discussions or questions.
> >
> > ### References
> > [1] Albini, Emanuele, et al. "Counterfactual shapley additive explanations." Proceedings of the 2022 ACM Conference on Fairness, Accountability, and Transparency. 2022.
> >
> > [2] Kommiya Mothilal, Ramaravind, et al. "Towards unifying feature attribution and counterfactual explanations: Different means to the same end." Proceedings of the 2021 AAAI/ACM Conference on AI, Ethics, and Society. 2021.

---

### Official Review · Reviewer_iyvf · 2024-11-07

**Soundness:** 3
**Presentation:** 3
**Contribution:** 3
**Rating:** 8
**Confidence:** 4

**Summary:**

The paper titled "Refining Counterfactual Explanations with Joint-Distribution-Informed Shapley Towards Actionable Minimality" proposes a framework for improving counterfactual explanations (CE) in machine learning models. Current CE methods often include unnecessary feature changes, making them difficult to apply in practical scenarios. The approach proposed in this paper seeks to minimize these changes while maintaining the validity of the counterfactual exercise, making CE more actionable for both users and stakeholders, particularly in terms of simplified and actionable decision-making.

*Major contributions of the paper include:*
1. The setup of a well-defined problem for finding CE with minimal feature changes (Equation 1).
2. The development of the algorithm "COunterfactuals with Limited Actions (COLA)" to find the solution to the previous point.

*Sub-contributions that are incorporated into the definition of COLA:*
- The definition of a generic Shapley framework (p-SHAP), which nests several other commonly used Shapley frameworks for computing feature importance and allows for incorporating the distributional connection between factual data $x$ and counterfactual data $r$, i.e., their joint distribution $Prob(x,r)$, in its computation.
- They show that p-SHAP correctly measures the causal behavior of shifting $x$ towards $r$ (Theorem 4.2).
They propose Optimal Transport (OT) techniques to recover $Prob(x,r)$, given $x$ and $r$, which can be incorporated into p-SHAP for CE. The OT guarantees tighter bounds on the distance between factual and counterfactual, $D(f(x), y^*)$, under the assumption that the model $f$ is Lipschitz continuous and $y^* = f(r)$ (Theorem 4.1).
- They derive the computational complexity of COLA.

The authors combine a collection of algorithms in order to construct COLA. Each of the algorithms is listed below:

a. FIND $r$: For a counterfactual $y^*$ and a factual $x$, find $r$ within an $\epsilon$ radius of $x$ to minimize the distance $D(y^*, f(r))$.

b. COMPUTE $Prob(r,x)$: Once $r$ is found, use it to estimate the joint density $Prob(x,r)$, (mostly) with OT.

c. COMPUTE p-SHAP values: Once $Prob(x,r)$ is obtained, use it to estimate p-SHAP values $\phi$ between $x$ and $r$.

d. OBTAIN THE CE: Using $\phi$, together with $Prob(x,r)$ and $r$, edit $x$ minimally in order to obtain the CE of $y^*$ from $x$, called $z$.

*Numerical Results:*
- The authors test their algorithm using four different datasets, where the task is classification. They show that the algorithm is feasible and achieves $z$ with empirically minimal changes compared to five other CE algorithms, with minimal or zero misalignment $D(f(z), y^*)$, across 12 different classifier models.
- In simpler setups, the authors compare COLA’s performance in terms of finding the $z$, measuring its performance $D(f(z), y^*)$, while comparing it to the optimal theoretical performance. Although not achieving optimality for some scenarios, COLA sometimes still outperforms CE methods that use exact alignment.

**Strengths:**

- The paper attempts to contribute to important theoretical explainability literature, in addition to connecting to real-world demands to make model explanations more actionable. This is a growing demand in several areas of society, especially with current and incoming AI regulation.
- The authors thankfully write for a broader audience without being too verbose. Although most of the paper’s contribution is presented starting in Section 4, the content before this section provides several points of entry and the exact amount of revision a reader needs to understand their framework. I personally learned from their exposition.
- The method is mostly sound and tries to be as generic as possible (e.g., as p-SHAP is a generic formulation), and the authors provide intuition for most of their steps. Overall, the paper seems well-polished and organized, lacking only a few important adjustments.
- A clear algorithm is defined, consisting of a collection of widely used algorithms, and its computational complexity is derived, which is informative for scalability.
- Their methodology is tested in a wide variety of settings, i.e., using four different divergence functions, four different datasets, comparing COLA to five other CE algorithms, and utilizing 12 different classifiers. This is useful to demonstrate the robustness of their method compared to others.

**Weaknesses:**

*PRESSING ISSUES*
- COLA has a well-defined procedure to compute $z$. It seems clear from Theorem 5.1 that $z$ would satisfy the constraints from Equation 1, but it is not clear how the chosen $z$ from COLA is actually optimal, i.e., why is it minimizing, or closer to minimizing, the problem in Equation 1, given that the last algorithm in COLA to find $z$ does not minimize any function. Even if it is not optimal, as suggested by Figure 4, the paper would benefit from having a clearer narrative of why COLA is closer than other methods to optimal, considering the minimization problem in Equation 1.
- The connection between $p-SHAP$ and $p_{OT}$ in Section 4 can be improved. This is the part that seems to be able to benefit the most from clarity and extra sentences connecting these two different topics and situating them in the bigger goals of the paper. A few suggestions:
    - When introducing the OT problem in this section, the reader might benefit from having a sentence or two mentioning what the goal of OT is, in addition to its purpose in p-SHAP. This is somewhat done in row 209, but I believe it would be clearer if it comes earlier in the previous paragraph.
    - It would be great to have a clear explanation/insight in the paper about why $A_{prob}$ does not necessarily require knowledge of how $r \sim \mathcal{D}$ is generated in order to achieve good CE performance.
    - In the “Theoretical Aspects of p-SHAP”, it would be helpful to know more precisely why using OT for obtaining the joint distribution $Prob(r,x)$ helps later with minimizing $D(f(z),y^*)$. This connection doesn’t seem to be precise as it is.
    - Theorem 4.1 is labeled as “p-SHAP towards CE”, but the theorem is much more a result of OT + Lipschitz continuity of $f$, and its direct connection to p-SHAP seems to be missing.

*OTHER*
- The paper provides so many entry points for the reader, and it would be nice to also have the definition of “Exact Alignment” and “Feature Alignment (FA) performance” at least once in a footnote.
- Perhaps writing a sentence at the beginning of Section 3, and/or at the end of Section 2, so the reader knows what to expect in the transition from Section 2 to Section 3, and how they fit in the big picture of the paper.
- Row 472: Result III could benefit from having a sentence entry point to situate the reader as to why this is important and how it fits in the big picture of the paper. Furthermore, another sentence would be useful to provide insights on why $CF-P_{OT}$ outperforms $CF-P_{Ect}$.
- Please add titles to the x-axes of Figures 3 and 4, either in the image or in their descriptions.

**Questions:**

Specific suggestions (please feel free to ignore these if I didn’t understand things well):
- In order to make this methodology widespread and for reproducibility, I don’t know if the authors considered converting their code (in the supplemental material) to a Python package or creating a Python class? I am not sure if the editor should require this for publication?
- Typo on row 196? “First p-SHAP degrades to RB-SHAP”, shouldn’t it be B-SHAP?
- Typo on row 474? Word alignment misspelled?
- Perhaps clarify on row 231 that it is the tightest Lipschitz upper bound, not necessarily the tightest upper bound.

---

> ### Author Response · Authors · 2024-11-14
>
> We sincerely appreciate the time and effort that the reviewer, iyvf, has invested in thoroughly reviewing our paper. We especially want to thank you for accurately summarizing our paper’s contributions and for providing such a detailed description of the key steps in COLA, as outlined.
>
> We are truely grateful for your recognition of the importance of the problem we are addressing and acknowledgment of our contributions on both the theoretical and empirical fronts. We are deeply thankful for your supportive comments and constructive insights.
>
> Below, we made every effort to address the points you raised. Please feel free to let us know if any aspects require further clarification. We remain open to any additional discussions you may have. (**Q** and **A** are questions raised by the reviewer and answers provided by the authors, respectively)
>
> ### Responses to Addressing “PRESSING ISSUES”
>
> **Q1:** “COLA has a well-defined procedure to compute $z$. It seems clear from Theorem 5.1 that $z$ would satisfy the constraints from Equation 1, but it is not clear how the chosen $z$ from COLA is actually optimal, i.e., why is it minimizing, or closer to minimizing, the problem in Equation 1, given that the last algorithm in COLA to find $z$ does not minimize any function. Even if it is not optimal, as suggested by Figure 4, the paper would benefit from having a clearer narrative of why COLA is closer than other methods to optimal, considering the minimization problem in Equation 1.”
>
> **A1:** Thank you for your insightful question. We acknowledge that the $\mathbf{z}$ obtained through COLA is not theoretically guaranteed to be a global optimum. However, $\mathbf{z}$ is aimed at minimizing the dissimilarity between $f(\mathbf{z})$ and the desired outcome $\mathbf{y}^*$.
>
> (**What function is COLA trying to minimize?** ) COLA refines $\mathbf{z}$ based on the feature attribution (FA) results in Lines 3-4 (as detailed in Algorithm 1 and Figure 2). Specifically, FA is used to determine the most important features of each data point in $\mathbf{x}$ that need to be modified in order to achieve the desired counterfactual effect $\mathbf{y}^* = f(\mathbf{r})$, while keeping the less important features unchanged. Theorem 4.1 shows that when optimal transport (OT) is employed to compute the joint distribution in $p$-SHAP, it minimizes an upper bound of the $1$-Wasserstein distance between $f(\mathbf{x})$ and $\mathbf{y}^*$. This implies that $p$-SHAP, during the FA process, identifies feature importance in a way that effectively moves $f(\mathbf{x})$ closer to $\mathbf{y}^*$ by modifying the relevant features of $\mathbf{x}$.
>
> In the revised version, we will add the following sentence before “Complexity of COLA” in line 349.
> > By Theorem 4.1, COLA aims to minimize the dissimilarity between $f(\mathbf{z})$ and $\mathbf{y}^*$ by modifying $\mathbf{z}$ based on feature attribution results, which identify the most important features to adjust to achieve the desired counterfactual effect.
>
> **Q2:** “The connection between $p$-SHAP and $p_{\text{OT}}$ in Section 4 can be improved. This is the part that seems to be able to benefit the most from clarity and extra sentences connecting these two different topics and situating them in the bigger goals of the paper. A few suggestions … ”
>
> **A2:** We completely agree with the reviewer on this point. Indeed, **our paper establishes a novel connection between the Shapley method and optimal transport**, which has not been previously reported. We appreciate your suggestions and will incorporate additional clarifications to strengthen this section. Below, we address your specific suggestions item by item.
>
> **Q3:** “When introducing the OT problem in this section, the reader might benefit from having a sentence or two mentioning what the goal of OT is, in addition to its purpose in p-SHAP. This is somewhat done in row 209, but I believe it would be clearer if it comes earlier in the previous paragraph.”
>
> **A3:** Agree, and thank you for the good suggestion! We will add the following sentences in the revised version when OT is introduced in Section 4, in the beginning of “Theoretical Aspects of $p$-SHAP”.
> >  Intuitively, OT determines the most cost-effective way to move $\mathbf{x}$ closer to $\mathbf{r}$. We later prove that the transportation plan $p_{\text{OT}}$, obtained by solving the OT problem, effectively guides $p$-SHAP in identifying the key features of $\mathbf{x}$ that need to be modified to bring $f(\mathbf{x})$ closer to $\mathbf{y}^*$.

---

> ### Author Response · Authors · 2024-11-14
>
> **Q4:** “It would be great to have a clear explanation/insight in the paper about why $A_{\text{Prob}}$ does not necessarily require knowledge of how $r\sim \mathcal{D}$ is generated in order to achieve good CE performance.”
>
> **A4:** Thank you for raising this very interesting point! To summarize, we would argue that not only does $A_{\text{Prob}}$ not require explicit knowledge of how $r\sim \mathcal{D}$ is generated to achieve good CE performance, but in fact, all components in COLA, except for $A_{\text{CE}}$, should exercise caution in utilizing knowledge of $r\sim \mathcal{D}$, where $\mathcal{D}$ is the distribution of counterfactual data.
>
> Let's revisit our core objective: The factual data $\mathbf{x}$, the model $f$, and the desired counterfactual effect $\mathbf{y}^*$ should all be determined in advance and remain fixed for our application, before choosing any counterfactual explanation (CE) algorithm to generate the counterfactual data $\mathbf{r}$ such that $\mathbf{y}^* = f(\mathbf{r})$. We want $A_{\text{Prob}}$ and $A_{\text{Shap}}$ to perform feature attribution (FA) for $\mathbf{x}$ and $f$ in order to achieve $\mathbf{y}^*$, and the FA results should be consistent and unique as long as $\mathbf{x}$, $f$, and $\mathbf{y}^*$ are fixed.
>
> However, if $A_{\text{Prob}}$ or $A_{\text{Shap}}$ depends on the distribution $\mathcal{D}$, then the results would no longer be unique. This is because, unlike $\mathcal{D}_{\text{Train}}$, which is observed from the real world, $\mathcal{D}$ is generated by a chosen CE algorithm.
>
> Since there are many CE algorithms available, each could produce a different $\mathcal{D}$ distribution, resulting in different FA outcomes for the same input data, the same trained model, and the same counterfactual effect. This variability would deviate from the fundamental objective of FA, which is to provide a unique and reliable explanation. The same reasoning applies to $A_{\text{Value}}$.
>
> We will add the following sentences in the revised version, after line 215.
>
> > The key reason that $p$-SHAP does not require explicit knowledge of how $r \sim \mathcal{D}$ is generated is because its goal is to work directly with the factual data $\mathbf{x}$, the model $f$, and the desired outcome $\mathbf{y}^*$, independent of the specific CE algorithm used to produce $\mathbf{r}$. By focusing solely on these fixed components, $p$-SHAP ensures consistency in FA without being influenced by the variability of different CE generation processes, which is a major difference to CF-SHAP.
>
> **Q5:** “In the “Theoretical Aspects of p-SHAP”, it would be helpful to know more precisely why using OT for obtaining the joint distribution $\text{Prob}(\mathbf{r},\mathbf{x})$ helps later with minimizing $D(f(\mathbf{z}), \mathbf{y}^*)$. This connection doesn’t seem to be precise as it is.”
>
> **A5:** Thank you! And we believe that the to-be-added sentence in A3 above should address this concern.
>
> **Q6:** “Theorem 4.1 is labeled as “$p$-SHAP towards CE”, but the theorem is much more a result of OT  + Lipschitz continuity of $f$, and its direct connection to $p$-SHAP seems to be missing.”
>
> **A6:** In the revised version, we will add the following sentence before “Complexity of COLA” in line 349.
>
> > By Theorem 4.1, COLA aims to minimize the dissimilarity between $f(\mathbf{z})$ and $\mathbf{y}^*$ by modifying $\mathbf{z}$ based on FA results, which identify the most important features to adjust to achieve the desired counterfactual effect.
>
> ### Responses to Addressing “OTHERS”
> **Q7:** “The paper provides so many entry points for the reader, and it would be nice to also have the definition of “Exact Alignment” and “Feature Alignment (FA) performance” at least once in a footnote.”
>
> **A7:** Thank you for your good suggestions! In the revised version, we will modify line 25 to be
>
> > … it may be misleading to rely on a counterfactual distribution defined by the CE generation mechanism ….
> This is because the exact alignment literally refers to a known distribution of a counterfactual, which is usually defined by a given CE method [1].
>
> Then, we will modify line 197 to be
>
> > …. when $A_{\text{Prob}}$ defines a joint distribution between $\mathbf{x}$ and $\mathbf{r}$, indicating an $i\leftrightarrow j$ alignment of any $\mathbf{x}_{i}$ and $\mathbf{r}_j$.
>
> This modification reinforces the concept that the distribution determines the alignment, as stated in line 308. And we will see if they need to be move to footnote based on the length of the paper after the draft revision.

---

> > ### Author Response · Authors · 2024-11-14
> >
> > **Q8:** “Perhaps writing a sentence at the beginning of Section 3, and/or at the end of Section 2, so the reader knows what to expect in the transition from Section 2 to Section 3, and how they fit in the big picture of the paper.”
> >
> > **A8:** Good advice! We plan to add the following sentences at the end of Section 2.
> > > To solve (1), we resort to FA to identify the most influential features to obtain the modification indicator variable $\mathbf{c}$. The next section introduces commonly used Shapley value methods for FA, which, together with our later proposed one, are integrated into our algorithmic framework COLA, to obtain the refined counterfactual $\mathbf{z}$.
> >
> > **Q9:** “Row 472: Result III could benefit from having a sentence entry point to situate the reader as to why this is important and how it fits in the big picture of the paper. Furthermore, another sentence would be useful to provide insights on why CF-$p_{\text{OT}}$ outperforms CF-$p_{\text{Ect}}$.”
> >
> > **A9:** Agree! To give an entry point to the reader to understand why Result III is important, we plan to add the following sentence at the beginning of Result III.
> >
> > > This result demonstrates the effectiveness of $p$-SHAP in eliminating the influence of the CE generation process by replacing the CE algorithm-dependent knowledge of $\mathcal{D}$ with the optimal transport (OT) joint distribution between the factual and counterfactual data, shown in Figure 4.
> >
> > This sentence reinforces the added content in A4 above.
> >
> > Then, we will add the following sentence in line 485.
> >
> > > CF-$p_{\text{OT}}$ outperforms CF-$p_{\text{Ect}}$ because it utilizes a more theoretically grounded approach to identify the key features that require modification, whereas CF-$p_{\text{Ect}}$ relies on CE algorithm-dependent knowledge, which lacks solid justification on its effectiveness for FA.
> >
> > ### Responses to “Questions”
> > **Q10:** “In order to make this methodology widespread and for reproducibility, I don’t know if the authors considered converting their code (in the supplemental material) to a Python package or creating a Python class? I am not sure if the editor should require this for publication?”
> >
> > **A10:** Thank you for bringing this up! Interestingly, we share the same thought as you. We have already developed an implementation of COLA as a Python software library (separate from the code used for the experiments in this paper, which is included in the supplemental material). However, since we are currently in the double-blind review process, we have decided to hold off on publishing this library for now. Once the review process concludes, we will promptly make it available on GitHub.
> >
> > **Q11:** “Typo on row 196? “First p-SHAP degrades to RB-SHAP”, shouldn’t it be B-SHAP?”
> >
> > **A11:** Sorry for this! It should be B-SHAP. We will fix it in the revised version.
> >
> > **Q12:** “Typo on row 474? Word alignment misspelled?”
> >
> > **A12:** Sorry again. It should be spelled as “alignment” rather than “alginment”. We will fix it in the revised version.
> >
> > **Q13:** “Perhaps clarify on row 231 that it is the tightest Lipschitz upper bound, not necessarily the tightest upper bound.”
> >
> > **A13:** Agree! To make the statement mathematically rigorous, we will clarify in row 231 that “…which in turn provides the tightest Lipschitz upper bound on …”.
> >
> > Once again, we sincerely thank the reviewer for dedicating so much time and effort to reviewing our paper! Your suggestions and insights have been **incredibly valuable**, and they **directly contribute to improving the quality** of our manuscript. We deeply appreciate your thoughtful feedback and remain open to further discussions if you may have additional points of interest or questions.

---

> > > ### Comment · Reviewer_iyvf · 2024-11-26
> > >
> > > Thanks for addressing these points. I really believe that it made your paper mode approachable for other researchers.

---

### Author Response · Authors · 2024-11-21
**Revised version uploaded**

Dear Reviewers,

Thank you for taking the time to review our manuscript. We have uploaded a revised version incorporating those modifications mentioned in our rebuttals. The revised contents are highlighted in red for your convenience.

We kindly ask for **your feedback** to confirm if our responses have addressed your concerns. We remain available for any further discussions or clarifications.

Best regards,
The Authors

---

### Meta-Review · Area_Chair_csFz · 2024-12-14

**Metareview:**

This paper presents a method to improve the "actionable efficiency" of counterfactual explanations – i.e., the number of features that we must change to attain a target prediction from a point $x_0$. This is a desirable property in applications where we may wish to provide recourse to decision subjects. In such applications, the vast majority of algorithms for recourse provision can yield an action plan $a_\textrm{CE}$ such that $f(x_0 + a_\textrm{CE}) = 1$. Using the proposed method, we can convert such plans into a simpler action plan $a_\textrm{COLA}$ such that $f(x_0 + a_\textrm{COLA}) = 1$ and $\|a_\textrm{COLA}\|_0 \leq \|a_\textrm{CE}\|_0$.

**Recommendation**

This paper received mixed scores in its initial reviews due to a number of concerns pertaining to clarity of exposition. The paper experienced a productive rebuttal period as authors provided clear explanations updated their manuscript. Following the rebuttal, the paper ended up with scores ranging from weak accept to accept. Given that reviewers with slightly more expertise assigned lower scores and lower confidence, I reviewed the paper carefully in preparing the meta-review -- reading the submission, the responses, and the authors' rebuttal.

Given my reading, my recommendation is to reject the submission at this time. In this case, my recommendation detracts from reviewers due to a number of issues that were missed in the review, and that would have a major impact on the soundness and significance of the contributions. I describe each issue in greater detail below. I recognize that the result may be disappointing given the positive feedback the authors received from reviewers at the end of the rebuttal period. I want to be clear that this is not a routine decision, and that I cannot think of a simple way to address these issues that would not require major revisions or repositioning.

**Major Issue 1 [Affects Soundness]**

One of the key issues in the paper is a failure to consider a broader class of actionability constraints (e.g., constraints to enforce monotonicity, integrality, preserve feature encoding, or ensuring causal dependencies). These constraints are not usually considered by work on feature importance and counterfactual explanations. In the context of this paper, however, they are relevant given the stated motivation to improve "actionable efficiency."

COLA has two specific issues because it cannot enforce neither enforce constraints not tell if they are satisfied. Given an action plan that obeys these broader actionability constraints, COLA can return an action plan that violates them -- leading a plan that is "sparse" but "infeasible." Given an action plan that violates actionability constraints, COLA may also fail to detect -- labelling it as "actionable" for users. In this case, my concerns are that:

[W1] The manuscript does not state these limitations (this is misleading, but could be addressed by explicitly stating limitations)
[W2] The severity of these issues are unknown (this is important, and could be studied through experiments).
[W3] The approach cannot be changed to address these limitations (I do not see how to fix this).

**Major Issue 2 [Affects Significance]**

One broader issue is that the weaknesses listed can be addressed using an approach presented in at ICLR last year (see [Kothari et al.](https://openreview.net/forum?id=SCQfYpdoGE&nesting=2&sort=date-desc)). This paper outlines a method to generate "action plans" that are model-agnostic, can enforce actionability constraints, and do not require post-processing the output of another algorithm. The method proposed in that paper must be extended slightly to enforce a sparsity constraint and handle continuous feature spaces – both of these appear to be feasible. On another note, the paper includes a Python library that the authors could use to study the severity of this problem (and handle [W1]-[W2]) as they also run experiments for heloc and German.

**Minor Issue 3 [Affects Novelty]**

One of the secondary contributions of this work is a series of findings that describe counterintuitive findings. Many of these point to trade-offs between feature attribution and counterfactual explanations, and overlap with recent theoretical and empirical results:
- [Bilodeau et al - Impossibility Results for Feature Attribution](https://arxiv.org/abs/2212.11870)
- [Fokuma et al – Attribution-based Explanations that Provide Recourse Cannot be Robust](https://arxiv.org/abs/2205.15834)
- [Cheon et al – Feature Responsiveness Scores](https://arxiv.org/abs/2205.15834)

**Additional Comments On Reviewer Discussion:**

See above.

---

### Decision · Program_Chairs · 2025-01-22

Reject